# Multiple convergent supergene evolution events in mating-type chromosomes

Sara Branco[1,2], Fantin Carpentier[1], Ricardo C. Rodríguez de la Vega [1], Hélène Badouin[1,3], Alodie Snirc[1], Stéphanie Le Prieur[1], Marco A. Coelho[4], Damien M. de Vienne[3], Fanny E. Hartmann[1], Dominik Begerow[5], Michael E. Hood[6] & Tatiana Giraud [1]

Convergent adaptation provides unique insights into the predictability of evolution and ultimately into processes of biological diversification. Supergenes (beneficial gene linkage) are striking examples of adaptation, but little is known about their prevalence or evolution. A recent study on anther-smut fungi documented supergene formation by rearrangements linking two key mating-type loci, controlling pre- and post-mating compatibility. Here further high-quality genome assemblies reveal four additional independent cases of chromosomal rearrangements leading to regions of suppressed recombination linking these mating-type loci in closely related species. Such convergent transitions in genomic architecture of mating-type determination indicate strong selection favoring linkage of mating-type loci into cosegregating supergenes. We find independent evolutionary strata (stepwise recombination suppression) in several species, with extensive rearrangements, gene losses, and transposable element accumulation. We thus show remarkable convergence in mating-type chromosome evolution, recurrent supergene formation, and repeated evolution of similar phenotypes through different genomic changes.

[1] Ecologie Systématique Evolution, Bâtiment 360, Univ. Paris-Sud, AgroParisTech, CNRS, Université Paris-Saclay, 91400 Orsay, France. [2] Department of Microbiology and Immunology, Montana State University, Bozeman, MT 59717, USA. [3] Univ Lyon, Université Lyon 1, CNRS, Laboratoire de Biométrie et Biologie Evolutive UMR5558, F-69622 Villeurbanne, France. [4] UCIBIO-REQUIMTE, Departamento de Ciências da Vida, Faculdade de Ciências e Tecnologia, Universidade NOVA de Lisboa, 2829-516 Caparica, Portugal. [5] Ruhr-Universitat Bochum, AG Geobotanik Gebaude ND 03/174 Universitatsstraße, 15044780 Bochum, Germany. [6] Department of Biology, University of Virginia, Gilmer 051, Charlottesville, VA 22903, USA. These authors contributed equally: Sara Branco, Fantin Carpentier, Ricardo C. Rodríguez de la Vega. These authors jointly supervised this work: Michael E. Hood, Tatiana Giraud. Correspondence and requests for materials should be addressed to T.G. (email: tatiana.giraud@u-psud.fr)

Gould's view that evolution is "utterly unpredictable and quite unrepeatable"[1] has long prevailed. It is difficult to test the repeatability of evolution, but such tests are essential for understanding biological diversification in response to selection[2,3]. Cases of convergent evolution following similar selective pressures provide ideal opportunities for assessing the repeatability of evolutionary processes and unraveling the proximal and ultimate mechanisms generating diversity[2,4]. Examples of convergent evolution include ecological morphs in Nicaraguan crater lake cichlid fishes[5], cave morphs in Mexican cavefishes[6], resistance to toxic compounds in animals[7], and lactase persistence in humans[8]. However, few examples have been studied in detail and many unresolved questions remain, including the frequency of convergent evolution, the genetic mechanisms underlying convergent trait evolution, whether convergence is widespread in organisms other than plants and animals, and the phylogenetic scales at which it occurs.

Supergenes (the beneficial linkage of genes controlling different ecological traits by recombination suppression) are striking cases of adaptation, arising by conspicuous changes in genomic architecture. As such, supergenes can be good models for assessing the predictability and proximate/ultimate causes of convergent evolution. Although we still know little about supergene prevalence and evolutionary importance[9–11], interesting cases have been reported, including the non-recombining genomic regions controlling multiple wing color patterns in butterflies[12] and polymorphic social behavior in ants[10,11]. Chromosomes involved in sexual compatibility often also have large non-recombining regions, and the early stages in development of these regions can be considered to constitute supergenes[13]. Recombination suppression linking different traits involved in sexual compatibility has been documented in the sex chromosomes of animals and plants[13,14], the mating-type chromosomes of algae[15] and fungi[16–21], and self-incompatibility loci in plants[22]. Recombination cessation not only maintains beneficial allelic combinations but also reduces selection efficacy, leading to genomic decay and the accumulation of transposable elements (TEs)[23]. The frequency and proximal mechanisms of recombination suppression and the tempo of genomic degeneration remain unclear[13,14,24]. Chromosomes with recent recombination suppression events are ideal for investigating the initial steps of supergene formation and degeneration[25–27].

Fungi provide excellent systems for studying the causes, consequences, and frequency of recombination cessation, as they have diverse mating-type-determining systems involving multiple genes[28] and mating-type chromosomes with recent recombination suppression events[16–21]. Most basidiomycetes (mushrooms, rusts, and smut fungi) have two independently segregating loci controlling mating type at the haploid stage: (1) the *PR* locus, containing a pheromone receptor gene and one to several mating pheromone genes involved in pre-fertilization compatibility, and (2) the *HD* locus, encoding two homeodomain transcription factors responsible for post-fertilization compatibility[28]. Linkage between *PR* and *HD* loci underlies a major transition determining reproductive compatibility in these fungi[17,20,29]. Such linkage was long thought to be rare but is beneficial in selfing mating systems[30,31] (Supplementary Fig. 1). Linkage between *PR* and *HD* loci results in large regions controlling multiple mating-type functions (pre- and post-fertilization compatibility), which can be described as supergenes, as they represent a beneficial allelic combination where linkage increases fitness.

Here we describe a remarkable case of multiple convergent events of beneficial linkage between the *PR* and *HD* mating-type loci, corresponding to the repeated formation of supergenes in multiple young mating-type chromosomes across closely related fungi. We studied species of anther-smut fungi (*Microbotryum violaceum* complex; Fig. 1), a group of selfing pathogens. The mating-type chromosomes in this group were first described in *Microbotryum lychnidis-dioicae*[32], in which the mating-type loci are linked by a large region without recombination resulting from

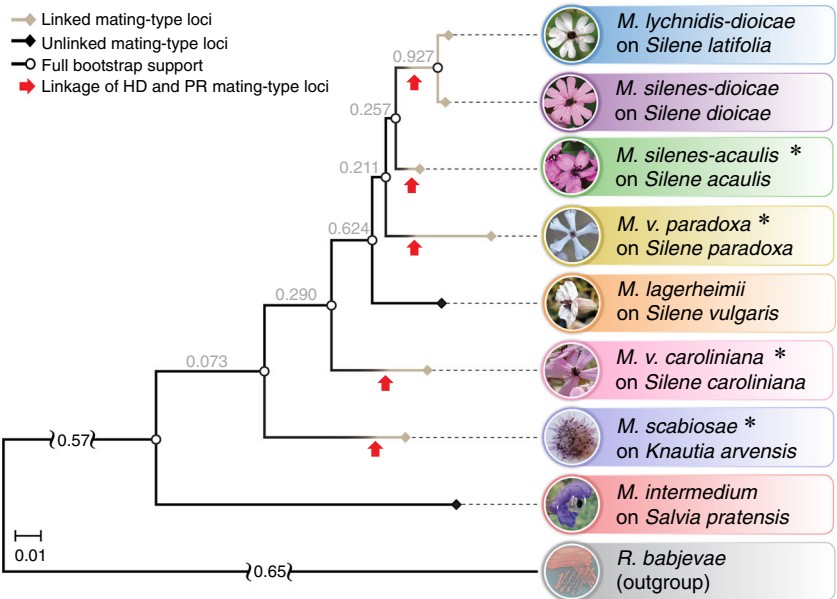

**Fig. 1** Phylogenies of anther-smut fungi and their breeding systems. Phylogenetic tree of the studied *Microbotryum* species (shown in the anthers of their host plants) and the outgroup *Rhodotorula babjevae*, based on 4229 orthologous genes. Species whose genomes were obtained in the present study are indicated by asterisks. Branch color and symbol indicate linked (gray branches and diamonds) or unlinked (black branches and diamonds) mating-type loci. The white circles indicate full bootstrap support. Red arrows indicate independent mating-type locus linkage events. Tree internode certainty with no conflict bipartitions (the normalized frequency of the most frequent bipartition across gene genealogies relative to the summed frequencies of the two most frequent bipartitions) is provided below the branches, indicating good support for the nodes. Relative certainty for this tree is 0.397

the fusion of the entire ancestral *PR* chromosome and one arm of the ancestral *HD* chromosome (Fig. 2)[16]. The two mating-type loci were precisely the ancestral limits of the initial recombination suppression event, with their linkage resulting in a supergene. Initial mating-type loci linkage was followed by further stepwise

expansions of suppressed recombination beyond mating-type genes[16]. These successive steps expanded the linked region and led to the formation of evolutionary strata with decreasing allelic divergence between mating-type chromosomes at increasing distance from the mating-type-determining genes[16], as observed

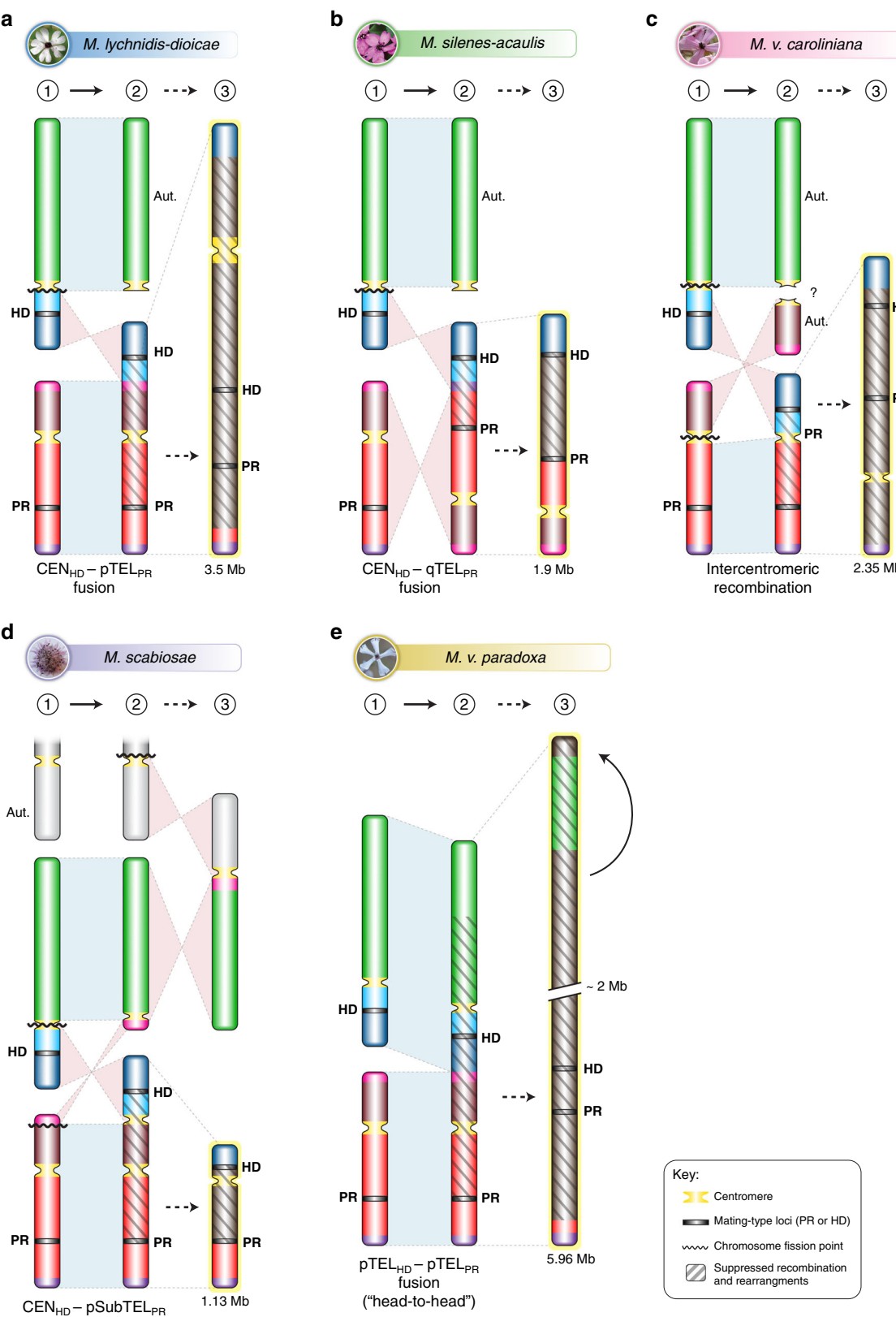

in animal and plant sex chromosomes[33] and likely other fungal mating-type chromosomes[17,18]. In contrast to the initial mating-type loci linkage event, these additional evolutionary strata did not control any traits for which linkage to mating-type would be beneficial[16]. Instead, the recombination suppression events occurring after mating-type locus linkage probably evolved to shelter deleterious alleles or through neutral rearrangements[16,24].

The majority of *Microbotryum* anther-smut fungi species display linked mating-type loci, but *M. lagerheimii* and *M. intermedium* have mating-type loci located on separate chromosomes[16,34] (Fig. 1). Given the phylogeny of anther-smut fungi (Fig. 1), a previous study based on parsimony inferred ancestral linkage between the *PR* and *HD* mating-type loci in the *Microbotryum* clade, with a reversal to unlinked mating-type loci in *M. lagerheimii*[34]. However, the distantly related species *M. lagerheimii* and *M. intermedium* have highly collinear mating-type chromosomes[16], whereas gene order is rearranged in non-recombining regions across species with linked *PR* and *HD* mating-type loci[16,19]. The collinearity between *M. lagerheimii* and *M. intermedium* mating-type chromosomes suggests that the ancestral state and gene order have been retained in these species. This raises the alternative hypothesis of a remarkable number of independent transitions linking the mating-type loci across anther-smut fungi (Fig. 1).

Using high-quality assemblies of closely related species and the ancestral gene order retained in *M. lagerheimii*[16], we uncovered four independent events of mating-type locus linkage in addition to the previously identified supergene[16]. The various *Microbotryum* species achieved mating-type locus linkage through different chromosomal rearrangements and have non-recombining regions of different sizes, ages, and gene contents. Our results show that supergenes can evolve frequently and that natural selection can repeatedly lead to similar phenotypes through multiple evolutionary trajectories and different genomic changes, consistent with repeatable evolution. We also document repeated and independent formation of evolutionary strata, with stepwise expansions of non-recombining regions beyond mating-type genes and provide evidence for increasing genomic decay in regions with a longer history of recombination suppression.

## Results

**Five independent routes for linking mating-type loci.** We inferred the evolutionary histories of mating-type chromosomes in multiple anther-smut fungi by comparing high-quality genome assemblies of eight *Microbotryum* species (Fig. 1). We obtained haploid genome assemblies for both mating types ($a_1$ and $a_2$) of four *Microbotryum* species with full linkage of *PR* and *HD* mating-type loci, as previously shown by progeny segregation[34]. We also studied available haploid genome sequences of four additional *Microbotryum* species[16] (Fig. 1; Supplementary Table 1). We used the *M. lagerheimii* genome as a proxy for ancestral gene order[16] due to its unlinked *PR* and *HD* loci and very few rearrangements relative to the distantly related *M. intermedium* species[16]. Whole-genome BLAST comparisons

revealed five different chromosomal rearrangements and fusions underlying the linkage between the *HD* and *PR* loci, one in each of the four newly assembled genomes and the fifth at the base of the previously analyzed clade containing *M. lychnidis-dioicae* and *M. silenes-dioicae*[16] (Figs. 1 and 2).

The different rearrangements led to variation across species in mating-type chromosome size and composition, as well as in non-recombining region length and captured gene content (Supplementary Table 1). Recombination ceased at different times, as shown by the different levels of synonymous divergence ($d_S$) between alleles associated with the $a_1$ and $a_2$ mating types for genes ancestrally located between the *HD* and *PR* loci (Figs. 3 and 4). For genes linked to mating-type loci, alleles associated with alternative mating types accumulate differences over time since the linkage event, whereas genes unlinked to mating-type loci are highly homozygous in these selfing fungi (with virtually no divergence between alleles present in the $a_1$ or $a_2$ haploid genomes within diploid individuals; Supplementary Figs. 2a–c). The absence of trans-specific polymorphism in genes ancestrally located between the *PR* and *HD* loci following chromosome fusion provided further evidence for the existence of five independent fusion events. Specifically, alleles of genes between mating-type loci clustered by species and not by mating type, demonstrating that their linkage to mating-type loci occurred after speciation events (Fig. 4; Supplementary Fig. 3). Only *M. lychnidis-dioicae* and *M. silenes-dioicae* displayed trans-specific polymorphism in the genomic regions ancestrally located between the *PR* and *HD* loci. Alleles associated with the $a_1$ mating type of both species consistently clustered together, as did alleles associated with the $a_2$ mating type, indicating that *PR*–*HD* linkage predated the speciation event in this clade (Fig. 4; Supplementary Fig. 3). Unlike mating-type chromosomes, autosomes were highly collinear between mating types and with no evidence of widespread interchromosomal rearrangements across species (Supplementary Fig. 4) or trans-specific polymorphism (only 1 of the 4229 single-copy shared autosomal genes displayed trans-specific polymorphism, and even then, only between *M. lychnidis-dioicae* and *M. silenes-dioicae*). The interchromosomal rearrangements and recombination suppression were thus restricted to the mating-type chromosomes and repeatedly led to regions of suppressed recombination bordered by the *HD* and *PR* loci. This indicates that the mating-type chromosome rearrangements linking mating-type genes were selected for, forming adaptive supergenes.

In *M. lychnidis-dioicae*, mating-type locus linkage was achieved by the fusion of the putative centromere end of the *HD* chromosome short arm to the distal end of the *PR* chromosome short arm (Fig. 2a). This evolutionary transition occurred before the divergence of *M. lychnidis-dioicae* and *M. silenes-dioicae*, as shown by their similar chromosome structures and trans-specific polymorphism, as previously reported[16]. Genealogies of genes ancestrally located between the *HD* and *PR* loci and calibrated using the date of speciation between *M. lychnidis-dioicae* and *M. silenes-dioicae*[35] indicated that the mating-type loci in this clade became linked 1.2 million years (MY) ago (Fig. 4). The timing of

**Fig. 2** Routes of mating-type chromosome evolution in *Microbotryum*. Model for mating-type chromosomal rearrangement events, as inferred from comparisons with the two mating-type chromosomes of *M. lagerheimii* (used as a proxy for the ancestral mating-type chromosomes in the genus[16]). Mating-type chromosome content across *Microbotryum* species is illustrated by colors referring to different parts of the two *M. lagerheimii* mating-type chromosomes (Supplementary Figs. 5–8). The inferred ancestral locations of putative centromeres and mating-type loci are indicated in yellow and black, respectively, and the regions of suppressed recombination are dashed. Chromosome sizes are indicated by their relative scales; the last stage in the evolution of recombination suppression often involves increases in chromosome size due to the accumulation of repetitive elements. Mating-type chromosome evolution in **a** *M. lychnidis-dioicae*, **b** *M. silenes-acaulis*, **c** *M. violaceum caroliniana*, **d** *M. scabiosae*, and **e** *M. v. paradoxa*, in which the top edge of the mating-type chromosome corresponds to a rearrangement from the middle of the chromosome, supporting complete recombination suppression up to the end of the chromosome

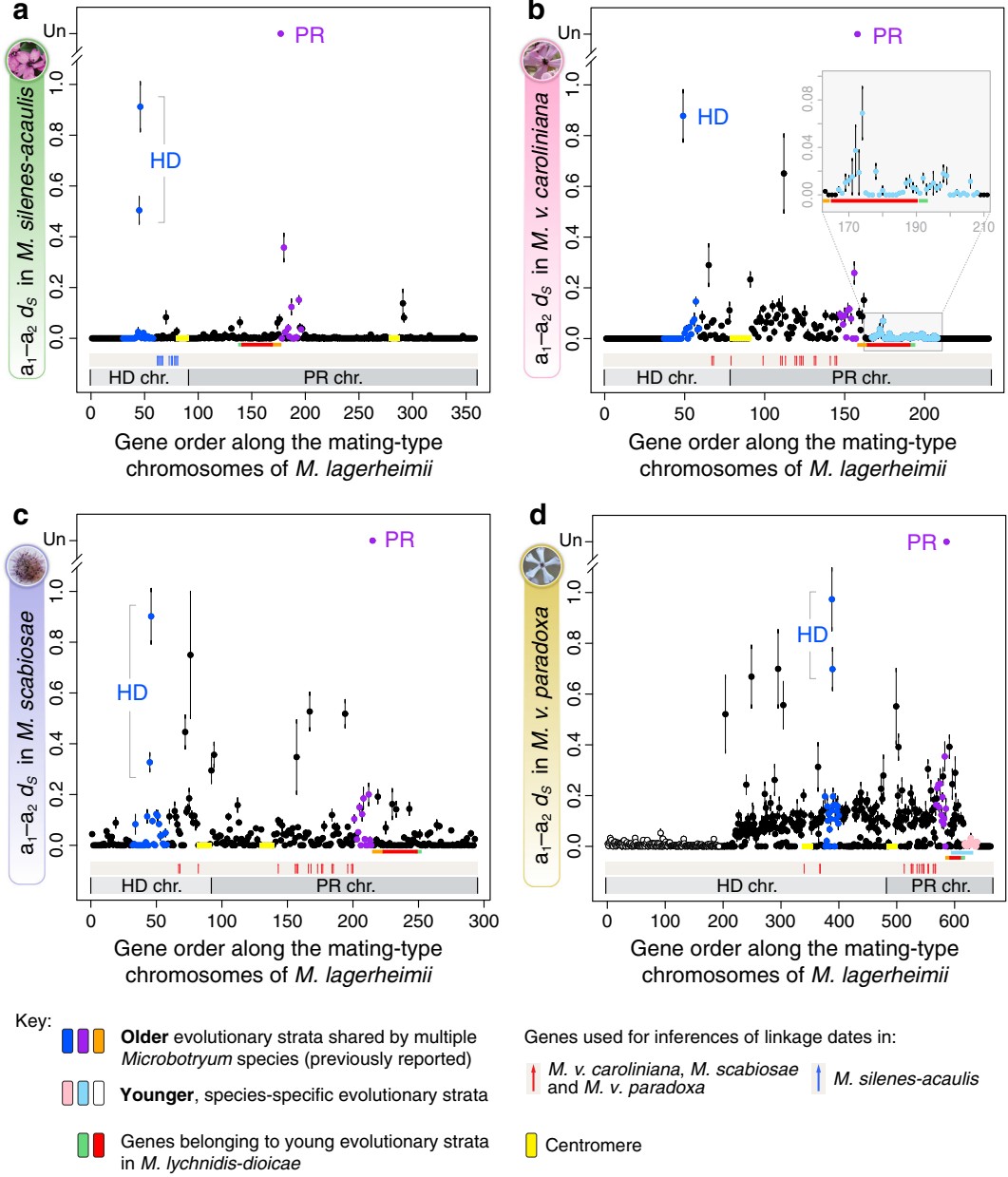

**Fig. 3** Divergence between a$_1$- and a$_2$-associated alleles. Per-gene synonymous divergence and standard error ($d_S$ ± SE) between alleles associated with the a$_1$ and a$_2$ mating types within *Microbotryum* diploid individuals, following the ancestral gene order for the mating-type chromosome. Synonymous divergence is plotted against the genomic coordinates of the a$_1$ mating-type chromosomes of *M. lagerheimii* for all single-copy genes common to both mating-type chromosomes. The limits of the PR and HD *M. lagerheimii* mating-type chromosomes are indicated and oriented according to the fusion in each species (i.e., not in the same orientation in all species). Divergence between the a$_1$ and a$_2$ pheromone receptor (PR) genes was too extensive and $d_S$ could not be calculated (depicted as "Un" for unalignable). The yellow boxes indicate the positions of *M. lagerheimii* putative centromeres. The red vertical arrows at the bottom indicate the 17 genes used for inferring HD–PR linkage dates in all species except for *M. silenes-acaulis*, for which we used a restricted set of 13 genes ancestrally located between the HD locus and the putative centromere (blue vertical arrows). Ancient evolutionary strata that evolved at the base of the *Microbotryum* clade are indicated in purple (around PR) and blue (around HD), as in the previous study in which they were discovered[16]. The genes involved in the more recent evolutionary strata previously identified in *M. lychnidis-dioicae*[16] are indicated with red and green bars at the bottom. **a** *M. silenes-acaulis*; **b** *M. v. caroliniana*, with a recent stratum indicated in light blue and enlarged in an inset; the current location of these genes is indicated in Supplementary Fig. 6a; **c** *M. scabiosae*; **d** *M. v. paradoxa*, with recent strata depicted in pink and white (the current location of these genes is indicated in Supplementary Fig. 8a). The light blue bar at the bottom indicates the genes involved in the young evolutionary stratum of *M. v. caroliniana*

*PR–HD* linkage was inferred from the divergence between alleles associated with the a$_1$ and a$_2$ mating types at 17 genes (red vertical arrows in Fig. 3). In these regions linked to mating-type loci, the alleles remained associated with the a$_1$ or a$_2$ mating type and diverged progressively with time since recombination suppression (Fig. 3). In *M. lychnidis-dioicae*, autosomes were

highly syntenic and without differentiation between alleles in the sequenced a$_1$ and a$_2$ genomes derived from a single diploid individual[19].

In *M. silenes-acaulis*, PR–HD linkage resulted from a different and more recent chromosomal rearrangement. As in *M. lychnidis-dioicae*, the short arm of the ancestral *HD* chromosome

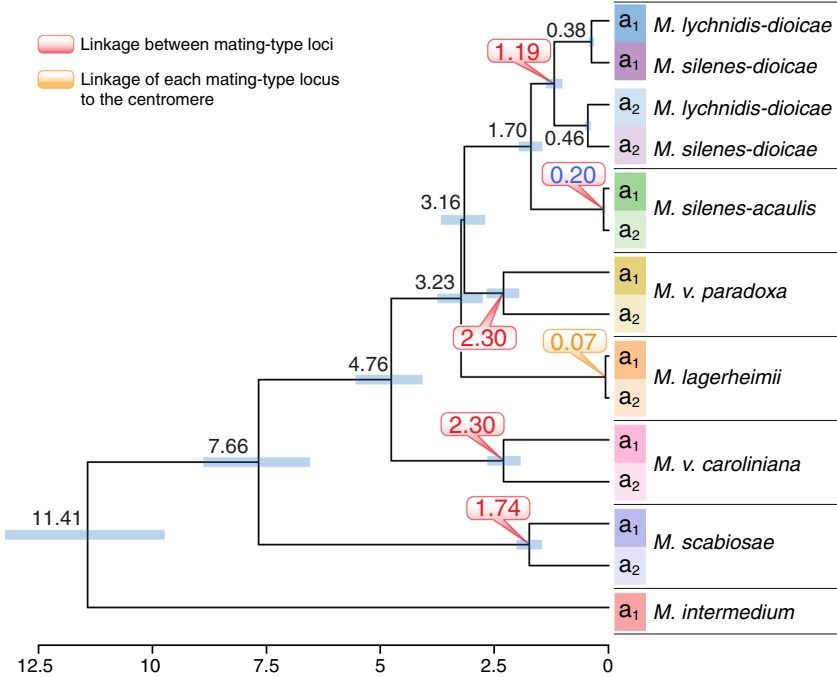

**Fig. 4** Dates of mating-type loci linkage. Reconstructed phylogenetic tree based on 17 concatenated genes ancestrally located between the mating-type loci after chromosomal fusion in all the studied species with linked mating-type loci but *M. silenes-acaulis* and including alleles from both $a_1$ and $a_2$ genomes. Numbers on tree nodes indicate the inferred dates of speciation (in black) and the events of mating-type loci linkage, either one to each other or to their respective putative centromeres (in red and orange, respectively). The blue bars correspond to 95% confidence intervals. The scale at the bottom indicates the time before present (million years). None of the individual genes showed trans-specific polymorphism, except between the sister species *M. lychnidis-dioicae* and *M. silenes-dioicae* (Supplementary Fig. 3). We used a restricted set of 13 genes (Fig. 3) for estimating the *M. silene-acaulis* mating-type loci linkage because not all 17 genes were located in its non-recombining region

fused with the entire ancestral *PR* chromosome, but the putative centromere end of the *HD* chromosome arm fused to the opposite end of the *PR* chromosome (Fig. 2b; Supplementary Fig. 5b, c). The much lower levels of synonymous divergence ($d_S$) between the alleles of genes positioned between the *PR* and *HD* mating-type loci (Fig. 3a; Supplementary Fig. 5a and Supplementary Table 2)[16] indicated a more recent fusion event. Because not all the 17 genes used for dating mating-type loci linkage in the other species were ancestrally located between the mating-type loci in *M. silenes-acaulis* (due to the PR fusion in the opposite direction in this species, Fig. 2), we used a restricted set of 13 genes ancestrally located between the *HD* locus and the putative centromere (blue vertical arrows in Fig. 3) for dating *HD–PR* linkage in this species. We estimated that mating-type loci linkage occurred ca. 0.2 MYs ago in *M. silenes-acaulis* (Figs. 3 and 4). Further evidence for a more recent chromosome fusion was also provided by the small number of rearrangements between the $a_1$ and $a_2$ mating-type chromosomes in *M. silenes-acaulis* (Supplementary Fig. 5a, b), contrasting with the extensive rearrangements observed in the *M. lychnidis-dioicae* non-recombining region[19]. In *M. silenes-acaulis*, a large inversion encompassing the region between the *PR* and *HD* loci (Supplementary Fig. 5a) may have contributed directly to the recombination suppression linking the two mating-type loci. All *M. silenes-acaulis* autosomes displayed high levels of synteny and $d_S$ values of zero between the sequenced $a_1$ and $a_2$ genomes originating from a single diploid individual (Supplementary Figs. 2a and 4a).

Other specific rearrangements led to mating-type locus linkage in the remaining species. All these rearrangements appeared older than those occurring in the ancestor of *M. lychnidis-dioicae* and *M. silenes-dioicae*, as shown by the higher $d_S$ levels (Fig. 3;

Supplementary Table 2) and the inferred earlier occurrence of recombination suppression based on the 17 gene set (Fig. 4). One of the oldest events was estimated to have occurred in *M. v. caroliniana*, about 2.3 MYs ago (Fig. 4). Unlike those of *M. lychnidis-dioicae* and *M. silenes-acaulis*, the *M. v. caroliniana* mating-type chromosome contained a single ancestral *PR* chromosome arm (Fig. 2c; Supplementary Fig. 6b)[16]. Higher $d_S$ values between the alleles of genes ancestrally positioned between the *PR* and *HD* loci (Fig. 3b, Supplementary Table 2) and massive rearrangements in the non-recombining region (Supplementary Fig. 6a) provided further evidence for an earlier onset of recombination cessation in *M. v. caroliniana* than in *M. silenes-acaulis*. All *M. v. caroliniana* autosomes were syntenic and with $d_S$ values of zero between the sequenced $a_1$ and $a_2$ genomes isolated from a single diploid individual (Supplementary Figs. 2b and 4b).

In *M. scabiosae*, a more recent event (1.7 MY old; Fig. 4) linked the mating-type loci following a chromosomal rearrangement similar to that in *M. lychnidis-dioicae* and *M. silenes-dioicae* (Fig. 2d) but with one extremity of the ancestral PR chromosome becoming incorporated into an autosome (black fragment in the outer track in Supplementary Fig. 7b; Fig. 2d). This particular configuration suggests ectopic recombination within a chromosome arm rather than rearrangement at putative centromeres as described above (Fig. 2). Consistent with the more recent recombination suppression, *M. scabiosae* displayed less extensive rearrangements between the $a_1$ and $a_2$ mating-type chromosomes than *M. v. caroliniana* (Supplementary Fig. 7a). Several large inversions nevertheless occurred between mating-type chromosomes. We were unable to sequence two meiotic products of a single diploid individual for *M. scabiosae*, which probably

explains the allelic variation observed between the sequenced $a_1$ and $a_2$ genomes even for pseudo-autosomal regions (PARs) and autosomes (Fig. 3c; Supplementary Figs. 2d and 7a). Nevertheless, the *M. scabiosae* mating-type chromosomes still appear to be exceptional in terms of rearrangements and divergence between the $a_1$ and $a_2$ genomes compared to autosomes (Fig. 3c; Supplementary Figs. 2d, 4c, and 7a).

Unlike those of all other species considered, *M. v. paradoxa* mating-type chromosomes resulted from the fusion of the entire ancestral *PR* and *HD* chromosomes (Fig. 2e; Supplementary Fig. 5k). This species experienced one of the earliest mating-type locus linkage events, with recombination suppression occurring about 2.3 MY ago (Fig. 4). This estimated age of recombination suppression is consistent with the high levels of rearrangements (Supplementary Fig. 8b) and high $d_S$ values (Fig. 3d) observed for *M. v. paradoxa* mating-type chromosomes. We obtained non-zero $d_S$ values across one side in most *M. v. paradoxa* autosomes, with zero $d_S$ values along the remaining length, as expected after an outcrossing event followed by a selfing event (Supplementary Fig. 2e). The $d_S$ values on autosomes remained much lower than those between the *HD* and *PR* loci on mating-type chromosomes (Fig. 3d; Supplementary Fig. 2e). There was also a very high degree of autosome synteny between the $a_1$ and $a_2$ genomes (Supplementary Fig. 4d), suggestive of ongoing recombination, as well as inter-species synteny, contrasting with the interchromosomal and intrachromosomal rearrangements observed for mating-type chromosomes (Supplementary Fig. 8a).

Gene genealogies provided further support for the existence of five independent mating-type locus linkage events. No transspecific polymorphism was found for any gene in the genomic regions ancestrally located between the *PR* and *HD* mating-type loci, other than in the sister species *M. lychnidis-dioicae* and *M. silenes-dioicae* (Fig. 4). Furthermore, with the exception of these two species, the inferred divergence date of alleles associated with the $a_1$ and $a_2$ mating types at genes ancestrally located between the mating-type loci following chromosome fusion was younger than at speciation events (Fig. 4).

Unlike all species described above, *M. lagerheimii* has unlinked mating-type loci (Fig. 1) despite also having a selfing mating system, as shown by the values of zero for $d_S$ obtained for all autosomes (Supplementary Fig. 2e). In this species, each mating-type locus is instead linked to the putative centromere of its chromosome[34], yielding the same odds of gamete compatibility as mating-type locus linkage under intra-tetrad selfing (Supplementary Fig. 1b, c). The linkage between the mating-type loci and putative centromeres was inferred to be very recent, occurring only ca. 0.07 MY ago (Fig. 4).

**Independent evolution of evolutionary strata**. Along with repeated and independent evolution of mating-type loci linkage by distinct genome rearrangements, we also observed the convergent evolution of subsequent expansion of the non-recombining regions forming evolutionary strata beyond the mating-type genes across multiple species. Such young evolutionary strata were defined as genomic regions with non-zero divergence between the alleles found in $a_1$ and $a_2$ genomes but with lower levels of differentiation than for the genomic region ancestrally located between the PR and HD loci. We identified these young evolutionary strata by plotting $d_S$ levels between the alternate alleles along the inferred ancestral mating-type chromosome gene order[16]. In organisms with high levels of selfing, such as *Microbotryum* fungi, $d_S$ is zero or very low in most diploid individuals (reflecting very high homozygosity levels, Supplementary Fig. 2). Non-recombining regions are a notable exception, where the degree of differentiation between alleles associated with the $a_1$ and $a_2$ mating types constitutes a proxy for time since linkage to

mating-type loci. Using this approach, we detected evolutionary strata extending recombination suppression beyond mating-type genes, including the two known ancient strata around the *HD* and *PR* loci common to all *Microbotryum* species (blue and purple strata[16], Fig. 3), as well as younger clade-specific strata. Some of the genes in the ancient (blue and purple) strata had low $d_S$ levels in some species, probably due to occasional gene-conversion events that reset the signal of divergence, as known to occur in fungal mating-type chromosomes[36,37].

We identified a young evolutionary stratum in *M. v. caroliniana* (light blue in Fig. 3b). This genomic region was located distally to the *PR* mating-type locus and had non-zero $d_S$ values significantly lower than the mean $d_S$ for genes ancestrally located between the *PR* and *HD* loci (Fig. 3d, Supplementary Table 2). The limit of the light-blue stratum was set at the most distal gene with a non-zero $d_S$ value, as all autosomes had zero $d_S$ values in the sequenced *M. v. caroliniana* diploid individual (Supplementary Fig. 2a). The light-blue stratum extended farther into the PAR than the most recent evolutionary strata in *M. lychnidis-dioicae* (red and green bars in Fig. 3b). The mean $d_S$ value in this evolutionary stratum was not significantly different from that in the PARs (Supplementary Table 2), indicating that mating-type locus linkage was recent. Such stretches of withinindividual non-zero $d_S$ genes were restricted to non-recombining regions in the *M. v. caroliniana* diploid individual sequenced (Supplementary Fig. 2b), providing strong evidence for recombination suppression in the light-blue region. Gene order in this region was largely conserved between the $a_1$ and $a_2$ mating-type chromosomes (Supplementary Fig. 5b), demonstrating that recombination can be halted in the absence of inversions. A small localized inversion within this stratum (orange links in Supplementary Fig. 5d) provided further evidence for the lack of recombination in this region. The autosomes were completely collinear between the two haploid genomes in the sequenced diploid individual (Supplementary Fig. 4b).

Evolutionary strata extending beyond the genes involved in mating-type determination were also detected in *M. v. paradoxa*. In this species, $d_S$ values were also highest in the nonrecombining region ancestrally located between the *PR* and *HD* loci (Fig. 3d; Supplementary Table 2) and were lower, but nonzero, in the two distal regions. These regions thus likely constitute two additional young evolutionary strata (white and pink, Fig. 3d). The region of recombination suppression extended farther into the PARs than in any of the other studied species, to the extent that only a single, very small PAR was retained (Fig. 3d and Supplementary Fig. 8a). We confirmed the complete suppression of recombination suppression in the white stratum by identifying a small region at the extremity of the *M. v. paradoxa* mating-type chromosome corresponding to rearranged genes ancestrally located in the center of the chromosome (Fig. 2e, and shown in gray in the outer track in Supplementary Fig. 8b). The pink region on the other side of the mating-type chromosome, distal to the *PR* locus (Fig. 3d), had high $d_S$ values but not higher than those of autosomes in the sequenced *M. v. paradoxa* individual and without inversions or rearrangements.

We confirmed recombination suppression in the pink, white, and light blue regions by sequencing multiple genomes for *M. v. caroliniana* and *M. v. paradoxa* (Supplementary Table 3). For genes linked to the mating-type loci, $a_1$- and $a_2$-associated alleles will differentiate to the point of forming two distinct clades in gene genealogies within species. Gene genealogies revealed such pattern for genes in the pink, white, and light blue regions, where the alleles associated with a given mating type were significantly more clustered than for genes in the PARs or in autosomes (Supplementary Fig. 9, Supplementary Table 4). In all three regions, all $a_1$ alleles branched in one clade and all $a_2$ alleles in

another clade in multiple gene genealogies (Supplementary Fig. 9), strongly supporting full linkage to mating type. Furthermore, the mean levels of polymorphism per mating type and per species were significantly lower in all the evolutionary strata than in the PARs (Supplementary Fig. 10, Supplementary Table 5), as

expected in regions without recombination due to the lower effective population size. These findings indicated that the non-zero $d_S$ values in the evolutionary strata within the sequenced individuals were due to recombination suppression rather than higher polymorphism levels.

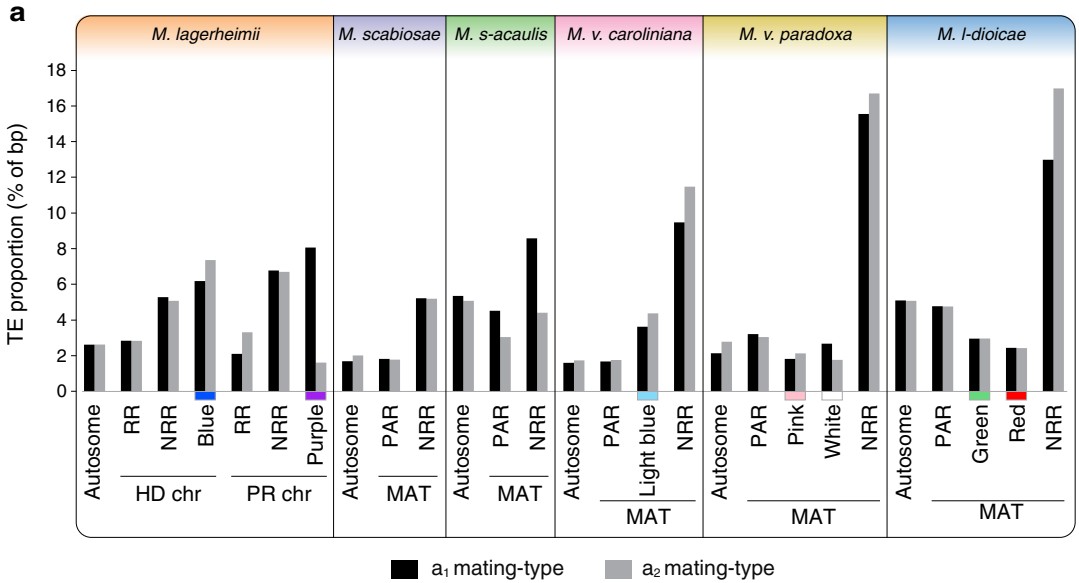

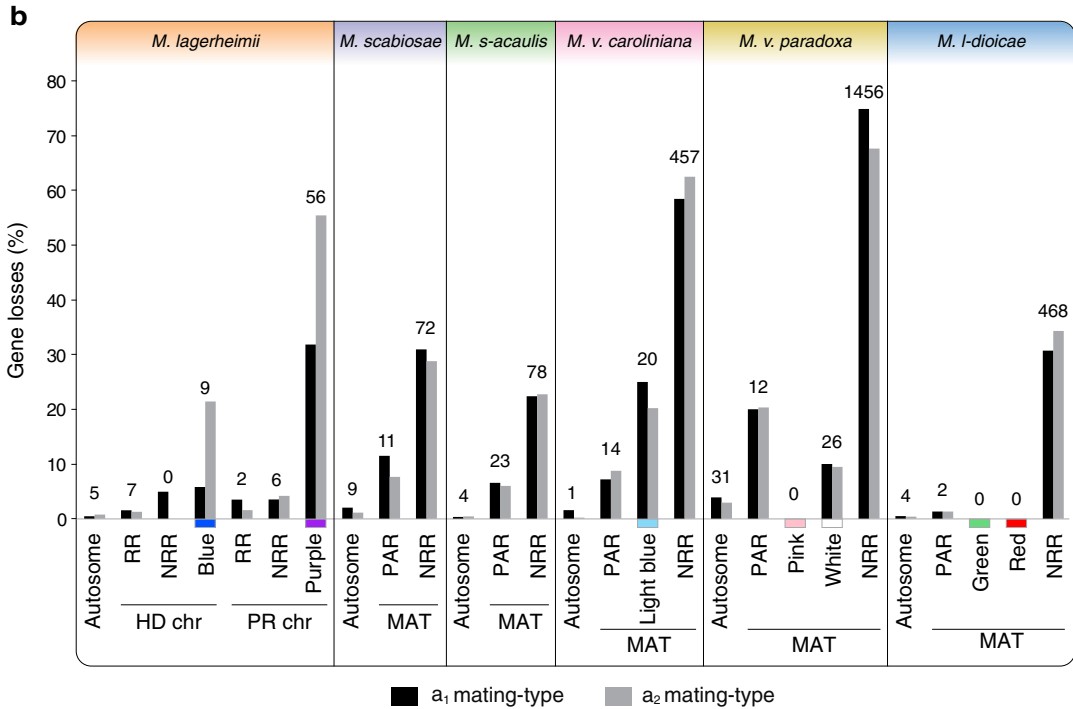

**Fig. 5** Differential degeneration across strata and species. We quantified the TE content and gene loss in genomes of both mating types (a₁ and a₂) of all species under study. For each species, we measured the TE accumulation separately for one fully assembled autosome (as a control), recombining regions (RR), and non-recombining regions (NRR) on mating-type chromosomes (MAT), separating the youngest evolutionary strata (light blue, red, green pink, and white strata) from the remaining of the NRR where applicable. Strata were ordered from the youngest to the oldest per species. In *M. lagerheimii*, the NRRs correspond to the regions between the mating-type loci and the putative centromeres, while in the other species they mostly correspond to the regions ancestrally between the HD and PR loci. The purple and blue strata were too rearranged within the large non-recombining region to quantify their specific gene loss or TE content except in *M. lagerheimii*. **a** Transposable element (TE) content (percent of base pairs); **b** Gene loss (genes with an allele present in the genome of one mating type but absent from the genome of the opposite mating type). Numbers at the top of the bars indicate the numbers of genes missing in the a₂ mating-type chromosome, present only in the a₁ mating-type chromosome

**Gene loss and TE accumulation**. We found evidence of differential gene loss and TE accumulation across species and evolutionary strata. Levels of gene loss and TE content increased with the age of recombination suppression, both within and across species (Fig. 5). Even the youngest evolutionary strata showed footprints of genomic decay relative to recombining regions (Fig. 5). The regions ancestrally located between the *PR* and *HD* loci displayed higher levels of degeneration in species with older recombination suppression events than in species with more recent mating-type locus linkage. Higher TE loads resulted in chromosomes larger than those in the ancestral state and contributed to differences in size between mating-type chromosomes (Figs. 2 and 5). The PARs and youngest strata displayed little evidence of TE accumulation compared to autosomes and showed low but non-negligible levels of gene loss (Fig. 5). Mating type ($a_1$ versus $a_2$) had no significant effect on gene loss or TE content, while differences between species were significant (Supplementary Table 6). The onset of genomic degeneration is thus rapid, with further gradual accumulation of TEs. Gene loss was extensive on both mating-type chromosomes: the two species with the most ancient recombination suppression between the mating-type loci lost between 60 and 70% of genes in this region within 2.3 MY, and *M. silenes-acaulis* has already lost >20% of genes within 0.20 MY.

## Discussion

We report an unprecedented case of convergent evolution, with five parallel recombination suppression events independently linking *PR* and *HD* mating-type loci in anther-smut fungi and generating megabase-long supergenes beneficial under selfing mating systems. We also reveal the convergent evolution of young evolutionary strata in multiple closely related species. Furthermore, our unique dataset suggests a progression of genomic decay in non-recombining mating-type chromosomes, with older regions of suppressed recombination displaying higher levels of gene loss and TE accumulation.

The convergent evolution of megabase-long supergenes in *Microbotryum* mating-type chromosomes, through distinct genomic rearrangements, has repeatedly led to the beneficial co-segregation of different mating-type functions (pheromones and pheromone receptors controlling pre-mating compatibility encoded by the *PR* locus, and the homeodomain proteins controlling post-mating compatibility encoded by the *HD* locus). A previous parsimony analysis indicated a very low probability of independent mating-type loci linkage events in anther-smut fungi and instead inferred reversal to the ancestral state of unlinked *PR* and *HD* loci in *M. lagerheimii*[34]. However, our analyses based on well-assembled chromosomes contradict these earlier inferences and reveal the striking occurrence of repeated convergent events linking mating-type loci through distinct genomic rearrangements. Further evidence for convergent linkage events is provided by differences in synonymous divergence between alleles associated with alternative mating types, the absence of trans-specific polymorphism in genes ancestrally located between the *PR* and *HD* loci, and differences in inferred linkage dates, with linkage occurring after speciation in at least five lineages. The existence of numerous other species with linked *PR* and *HD* mating-type loci across the *Microbotryum* genus[34,38] suggests the existence of many more independent events of mating-type locus linkage and convergent supergene formation in this genus. The occurrence of recombination suppression linking mating-type loci to each other or to centromere has been documented in a few interspersed lineages across fungi[17,18,21,29,31], but we provide here the first report of repeated convergent mating-type locus linkage events in multiple closely related species.

Repeated evolution of mating-type locus linkage in multiple closely related species implies very strong selection, which is supported by the lack of similar large-scale rearrangements or recombination suppression in autosomes. In selfing-based mating systems, as observed in all species of *Microbotryum* studied to date[39,40], *HD–PR* linkage increases the odds of compatibility between the gametes of a given diploid individual (Supplementary Figs. 1a, c) and is expected to be favored by selection. Interestingly, although the selfing species *M. lagerheimii* has unlinked mating-type loci[34], the *HD* and *PR* loci are linked to the putative centromeres of their corresponding chromosomes[34], which also increases gamete compatibility under selfing via intra-tetrad mating (automixis) (Supplementary Fig. 1b).

Our study provides invaluable insight into the frequency and importance of supergene evolution, a topic currently under explored[9]. Other examples of supergenes include the linkage of genes involved in mating types in plants and algae, mimicry wing patterns in butterflies, and complex social behavior in ants[10,11,13,15,22]. The repeated convergent evolution of non-recombining regions in anther-smut fungi provides strong support for the view that chromosomal rearrangements and the formation of supergenes are frequent and play an important role in evolution and adaptation[10,13,41]. The existence of repeated transitions in the genomic architecture of mating-type determination, following different chromosomal rearrangements, illustrates the power of natural selection and high genomic fluidity in shaping adaptation. Our findings suggest that natural selection drives evolution along trajectories leading to similar phenotypic outcomes through different genomic changes[2,4]. Very few instances of convergent evolution have been documented beyond the textbook examples of Nicaraguan crater lake cichlid fishes[5], cave morphs in Mexican cavefishes[6], and non-recombining chromosomes controlling polymorphic social behavior in ants[10,11].

We also found striking convergence in the stepwise extension of recombination suppression beyond mating-type genes, generating independent evolutionary strata in several *Microbotryum* species. The evolutionary causes leading to such strata devoid of mating-type genes are more likely to be the sheltering of deleterious alleles or neutral rearrangements than beneficial gene linkage[16,24,42]. Finding independent evolutionary strata in multiple closely related species provides further support for the occurrence of repeated evolution towards similar chromosomal states.

Our study provides important clues to the proximal mechanisms underlying the evolution of recombination suppression. Contrary to the common view that chromosomal inversions play a major role in preventing recombination[14], we found that recombination cessation can occur with the conservation of collinearity, as previously documented in fungi[16,20,21]. The occurrence of mating-type locus linkage via different routes reveals a high degree of genomic fluidity. Chromosomal rearrangements occurred frequently at putative centromeres, as in a recently reported case in the human fungal pathogen *Cryptococcus neoformans*[20]. These repeat-rich regions are highly labile and some of the inferred fusions in anther-smut fungi encompassed two ancestral putative centromeres in the same chromosome, as in *M. v. paradoxa*.

By the examination of repeated supergene formation events of contrasting ages, our results add further insights on the tempo of genomic decay and TE accumulation after recombination suppression. We found that gene loss occurred more rapidly than rearrangements or repeat accumulation. TE accumulation rates differed between species, probably because the *Microbotryum* species-specific TE loads[43] affect transposition rates. The high levels of degeneration observed in the two mating types probably

resulted from less efficient selection, due to lack of recombination and the sheltering of deleterious mutations in a permanently heterozygous state, with only very brief periods of haploid selection restricted to the meiotic tetrad stage[39]. Gene loss in these fungal mating-type chromosomes was more rapid than in the Y chromosome of the plant *Silene latifolia*[44], likely because plant sex chromosome degeneration is delayed by haploid purifying selection, unlike in animals or *Microbotryum* fungi[23]. Contrasting with sex chromosomes, genomic degeneration in *Microbotryum* mating-type chromosomes was not asymmetric (with no significant effect of mating-type on TE content or gene loss), as expected for organisms with an obligate heterozygous mating-type or sex chromosomes[45,46]. Lethal alleles linked to mating type were found at relatively high frequencies in natural populations of several *Microbotryum* species[47], preventing haploid growth in vitro but maintained through high levels of intra-tetrad mating. We also detected non-negligible levels of gene loss in the PARs. This may reflect the existence of very recent, undetected evolutionary strata or lower rates of recombination in the PARs compared to fully recombining autosomes[48]. This second hypothesis is consistent with the suggestion that partial deleterious allele sheltering in the PARs may account for evolutionary strata[16,24,42]: low recombination rates in the PARs would allow the accumulation of deleterious alleles, leading to selection for further recombination suppression and the permanent sheltering of these deleterious alleles.

In conclusion, our findings reveal remarkable repeated convergence in young mating-type chromosomes in closely related species, with supergenes evolving rapidly and frequently. Furthermore, our study shows that natural selection can repeatedly lead to similar phenotypes through multiple different evolutionary trajectories and genomic changes, rendering evolution predictable. The very recent advances in sequencing technologies yielding high-quality genome assemblies are allowing in-depth studies of chromosomal architecture and documenting the importance and prevalence of supergenes[9], as well as the high degree of genomic fluidity and convergence[4]. Future studies will certainly lead to the identification of many more cases of beneficial gene linkage, as predicted from evolutionary theory[41].

## Methods

**Strains, DNA extraction, and sequencing.** *Microbotryum violaceum* is a plant pathogen species complex that includes recently recognized cryptic and host specialized species, which have not all been formally named yet. For species without Latin names, we used *M. violaceum* (the terminology used to denote the whole species complex) followed by the name of the host plant, as is typically done in phytopathology for host races or *formae speciales*. We isolated a₁ and a₂ haploid cells from the following species: *M. violaceum caroliniana* parasitizing *Silene caroliniana* (strain 1250, Virginia Beach, USA, GPS Coord.: 36°54′36.0″N 76°02′24.0″W), *M. violaceum paradoxa* parasitizing *Silene paradoxa* (strain 1252, near Florence, Italy, GPS Coord.: 43°32′35.7″N 11°21′35.1″E), *M. silenes-acaulis* parasitizing *S. acaulis* (strain 1248, La Grave, France, GPS Coord.: 45°01′32.9″N 6°16′22.9″), and *M. scabiosae* parasitizing *Knautia arvensis* (strain 1118, Vosges, Retournemer lake, near Colmar, France, GPS Coord.: 48°03′00.0″N 6°59′00.0″E). The a₁ and a₂ haploid cells were isolated from single tetrads using micromanipulation for all strains except *M. scabiosae*, in which they were isolated from different teliospores.

DNA was extracted using a Carver hydraulic press (reference 3968, Wabash, IN, USA) for breaking cell walls and the Qiagen Anion-exchange columns Ref 10243 together with the buffers Ref 19060 (Courtaboeuf, France) for purifying DNA while avoiding fragmenting DNA. Haploid genomes were sequenced using the P6/C4 Pacific Biosciences SMRT technology (UCSD IGM Genomics Facility La Jolla, CA, USA).

**Assembly and annotation.** Genome assemblies were generated with the wgs-8.2 version of the PBcR assembler[49] with the following parameters: genomeSize = 30000000, assembleCoverage = 50. Assemblies were polished with quiver algorithm of smrtanalysis suite 2.3.0 (https://github.com/PacificBiosciences/

GenomicConsensus). A summary of raw data and assembly statistics for mating-type chromosomes is reported in Supplementary Table 1.

The protein-coding gene models were predicted with EuGene[50], trained for *Microbotryum*. Similarities to the fungal subset of the uniprot database plus the *M. lychnidis-dioicae* Lamole proteome[19] were integrated into EuGene for the prediction of gene models.

Mating-type chromosomes were identified by: (1) identifying the contigs carrying the PR and HD mating-type genes, (2) blasting the a₁ against the a₂ haploid genomes and visualizing the output using Circos[51] to identify contigs lacking collinearity, (3) blasting the haploid genomes against the completely assembled mating-type chromosomes of *M. lychnidis-dioicae*[19] and *M. lagerheimii*[16], (4) blasting the identified a₁ contigs to the whole a₂ haploid genome, and vice-versa, to detect which additional alternative mating-type contigs were linked to the previously identified mating-type contigs, and (5) re-doing steps (3) and (4) until no additional contig was identified. These contigs were then orientated in comparison to each other by: (1) using the putative centromere-specific repeats, as initial assemblies often yielded chromosome arms broken at the putative centromeres with identifiable putative centromere-specific repeats on each separated contig (e.g., Supplementary Figs. 5–8), and (2) blasting the a₁ and a₂ mating-type contigs against each other for identifying the PAR as the collinear regions that were then assigned to the edges of the chromosomes. The center contigs without centromeric repeats at any of their edges could not be oriented and were plotted in an arbitrary orientation.

**Orthologous groups, species tree, and $d_S$ plots.** To study the evolution of suppressed recombination in a phylogenetic context, we reconstructed the relationships between the nine *Microbotryum* species and a closest outgroup (*Rhodotorula babjevae*) for which genomes were available. The genomes of these species were either sequenced for this study or obtained from previous studies[16,19] (Fig. 1). A previously published genome of *M. intermedium* was used from a strain collected on the plant *Salviae pratensis*, while its usual hosts belong to Dipsacaceae; species identity has, however, been double-checked using ITS sequences. We compared the translated gene models of the *Microbotryum* species and the outgroup with blastp 2.2.30+. The output was used to obtain orthologous groups by Markov clustering as implemented in orthAgogue[52]. We aligned the protein sequences of 4229 fully conserved single-copy genes with muscle v3.8.31[53] and obtained the codon-based CDS alignments with TranslatorX[54]. We used RAxML 8.2.7[55] to obtain maximum likelihood gene trees for all fully conserved single-copy genes and a species tree with the concatenated alignment of 2,172,278 codons with no gaps (trimal -nogaps option) under the GTRGAMMA substitution model. We estimated the branch support values by bootstrapping the species tree based on the concatenated alignment and by estimating the relative internode and tree certainty scores based on the frequency of conflicting bipartitions for each branch in the species tree among the fully conserved single-copy genes[56].

For $d_S$ plots, we identified alleles using orthologous groups with a single sequence in each haploid genome for a given species. We used MUSCLE[53] embedded in TranslatorX[54] to align the two alleles per gene per species. Synonymous divergence and its standard error were estimated with the yn00 program of the PAML package[57].

**Figures and statistical tests.** Supplementary Figs. 4–8 were prepared using Circos[51]. We analyzed gene order after removing TEs to identify larger blocks of synteny. We identified syntenic blocks by searching all one-to-one gene correspondences between pairs of haploid genomes based on the orthologous groups reconstruction (see above). Statistical tests (Student's *t*-test, analysis of variance, and Wilcoxon rank tests) were performed using JMP v7 (SAS Institute).

**Date estimates for recombination suppression and genealogies.** For dating HD/PR linkage, we used alignments including a₁- and a₂-associated alleles at 17 single-copy orthologous groups that were located between the PR and HD loci following chromosomal fusion in all species but *M. silenes-acaulis* and that had both alleles retained (red vertical arrows in Fig. 3). The divergence between alleles associated with the a₁ versus a₂ mating types in these genes corresponds to the date of their linkage to mating-type loci. Indeed, genes linked to mating-type loci maintain one allele associated with a₁ and another allele associated with a₂ over time and the differentiation between these two alleles increases with the time since linkage to mating type. We only used the genes that displayed trans-specific polymorphism between *M. lychnidis-dioicae*, *M. silenes-dioicae* and *M. violaceum sensu stricto*[16] to avoid biasing estimates to younger dates because of gene conversion. For *M. silenes-acaulis*, we used a restricted set of 13 genes ancestrally located between the *HD* locus and the putative centromere (blue vertical arrows in Fig. 3a) because most genes located in its non-recombining region were in recombining regions in other species (Fig. 2). Divergence times were estimated using BEAST v2.4.0[58], with XML inputs generated using BEAUTi, and the default parameters except for unlinked substitution (HKY+G with empirical frequencies for each codon position) and clock models, Yule process to model speciation, and 10,000,000 mcmc generations sampled every 1000. We used a single calibration prior at 0.42 MY for all runs, corresponding to the divergence between

*M. lychnidis-dioicae* and *M. silenes-dioicae*[35], with a normal distribution and a sigma of 0.05. In some of the 17 individual gene trees ancestrally located between the HD and PR loci, some basal nodes were different from those in the species trees (Supplementary Fig. 3). However, the incongruent nodes were weakly supported so that gene genealogies were actually not significantly different from the species tree ($P > 0.35$, AU test[59]). We therefore forced the tree resulting from the concatenation of these 17 genes to the species tree topology for the date estimate analysis in BEAST. Genealogies of these 17 genes were inferred for codon-based alignments of genes in the different strata using RAxML[55] version 8.2.7, assuming the GTRGAMMA model and rapid bootstrap (options: -f a and -# 100).

**Identification of TEs.** Repetitive DNA content was analyzed with RepeatMasker[60], using REPBASE v19.11[61]. We used the RepeatMasker output to compute the percentage of base pairs occupied by TEs across the different evolutionary strata and PARs. For these counts, putative centromeres were filtered out. For plotting $d_S$ along chromosomes, repeats were removed. Further filtering of repeats was performed by blasting (tBLASTx), with removal of repeats matching to more than five locations in the genome.

**Detection of centromeric repeats.** We identified centromeric-specific repeats using a method specifically designed for this purpose[62], based on the observation that in most species studied to date putative centromeres contain the most abundant tandem repeats, are gene poor, and repeat rich. For identifying centromeric repeats, we used Tandem-Repeat Finder (TRF v. 4.07b[63]) on assembled Illumina reads of the *M. lagerheimii* strain as the one sequenced using the Pacific Bioscience technology. We performed the assemblies as follows: we randomly chose 500,000 Illumina reads that we assembled with PRICE v1.2[64] using a random set of 1,000,000 reads as seed file and using the following command line arguments: -fpp (or -mpp when using mate-pair reads) inputFile_R1 inputFile_R2 650 90 -picf 20000 seedFile 500 2 25 -nc 10 -mpi 85 -MPI 95 – tpi 85 -TPI 95 -logf logfile -o outputFile. PRICE works by rounds of assembly: in the first round, it maps randomly picked reads onto contigs (provided by the "seedFile), assembles the reads that did not mapped, and then extends the contig with the unmapped assembled sequences. For the second and following rounds, PRICE considers the extended contigs as the reference to restart the process of picking, mapping reads, assembling the unmapped reads, and extending the reference contigs. We analyzed the presence of tandem repeats in each of the 10 assembly cycle outputs using the following parameters in a TRF wrapper perl script[62]: match = 1, mismatch = 1, indel = 2, probability of match = 80, probability of indel = 5, min score = 200, max period = 2000. We performed these steps 15 times, picking randomly 500,000 input reads and 1,000,000 reads for the seed file. The repeats detected in the Illumina genomes were blasted against the corresponding high-quality genomes. We identified the putative centromeres in *M. lagerheimii* as the most gene-poor and repeat-rich regions and with the most abundant tandem repeats. The delimitations of the centromeric regions using this method yielded a single region per contig and were congruent with those using BLAST of the centromeric repeats identified previously in *M. lychnidis-dioicae*[19]. Putative centromeres were identified in the other species by blasting the identified centromeric-specific repeats (the tandem repeats identified here and previously[19] gave congruent results, see Supplementary Figure S4b).

**Gene loss.** Alleles were identified by applying orthomcl[65] to the protein data sets for unique $a_1$ and $a_2$ orthologs, discarding orthologous groups containing more than one protein-coding gene per mating type. The loss of a gene was inferred when a protein-coding gene in one mating type did not have any match in the orthomcl output in the opposite mating type within a diploid genome. We computed the number of gene losses across the PARs and the evolutionary strata that were not too rearranged to be delimited.

**Polymorphism data and analyses.** To rule out high levels of polymorphism as the cause for the observed high $d_S$ values in the youngest strata of the *Microbotryum* mating-type chromosomes, we assessed the level of polymorphism and $a_1$–$a_2$ allelic segregation in gene genealogies. When genes are linked to mating-type loci, alleles associated with the $a_1$ versus $a_2$ alleles accumulate differences until completely segregating according to mating-type allele rather than according to strain in gene genealogies. To test for this pattern, we re-sequenced multiple strains of *M. v. paradoxa* and of *M. v. caroliniana* (4 strains for *M. v. paradoxa* and 11 strains for *M. v. caroliniana*, Supplementary Table 3, strains collected before 2014 and thus not falling under the Nagoya protocol) from $a_1$ and $a_2$ haploid sporidia isolated from single tetrads using micromanipulation. For *M. v. caroliniana*, we also used spores collected from *S. virginica*, as this plant species is parasitized by the same anther-smut species as the one parasitizing *S. caroliniana*. Haploid sporidia were cultured on potato dextro agar and DNA was extracted using the Nucleospin Soil Kit (Macherey-Nagel, Germany). Haploid genomes of identified mating type were sequenced (Illumina paired-end sequencing with 46× mean coverage).

After trimming and filtering for quality (length >50; quality base >10) using cutadapt[66], reads were mapped against the high-quality PacBio reference genome of the same mating type and species. We used bowtie2[67] in the "very-sensitive-local" mode with the default parameters. Mapped reads were filtered for PCR

duplicates using picard-tools (http://broadinstitute.github.io/picard) and realigned on the PacBio reference genome using GATK[68]. Single-nucleotide polymorphisms (SNPs) were called with GATK HaplotypeCaller, which provide a gVCF per strain. For each strain, we filtered on a quality above 100 and other parameters (QD, FS, MQ, MQRankSum and ReadPosRankSum) for which the threshold was the fifth percentile (95th for FS parameter). The subsequent SNPs having >90% of missing data among strains were excluded from the dataset.

We generated pseudo-sequences per species and mating type by substituting reference nucleotides by their variants in the reference sequence, using the predicted CDS of the PacBio reference genome and the VCF file produced by GATK GenotypeVCF that combine gVCF into one file. We then computed the θπ statistic of diversity with EggLib version 2[69] for each species and mating type.

We generated alignments of $a_1$ and $a_2$ alleles per species and per predicted coding sequence and with $a_1$ and $a_2$ alleles from the predicted orthologous genes of the *M. lagerheimii* reference genome. Codon-based alignments were performed for each single-copy gene in the mating-type chromosome, for the various evolutionary strata, and the pseudo-autosomal regions, as well as for a well-assembled autosome, using TranslatorX[54]. Trees were computed with the RAxML rapid bootstrap mode (-f a -m GTRGAMMA -# 100) and plotted and rooted on the *M. lagerheimii* $a_1$ strain using the R ape package[70].

**Clustering of mating-type-associated alleles in genealogies.** For genes linked to the mating type loci, $a_1$- and $a_2$-associated alleles will differentiate to the point of forming two distinct clades in gene genealogies. To assess the level of clustering of alleles retrieved from $a_1$ and $a_2$ genomes in the gene genealogies, we designed and computed the following index: for each tree, we successively sampled the closest pairs of $a_1$- and $a_2$-associated alleles—in terms of nodal distance—until no pair was left (leaving out singletons in cases where the numbers of $a_1$ and $a_2$ alleles were not identical). This was performed 10 times in order to remove a possible effect of the order in which the pairs were sampled. We then computed the mean of these minimum values and compared it to a null distribution of the same index obtained by randomly permuting 1000 times the leaves of the input tree. The index was defined as the proportion of the random permutations giving a mean minimum value smaller or equal to the observed one. Index values close to 1 mean $a_1$ and $a_2$ alleles were completely separated in the tree (i.e., permutations always brought $a_1$- and $a_2$-associated alleles closer than they actually were). Conversely, an index value close to 0 meant that the $a_1$- and $a_2$-associated alleles were all forming pairs ("cherries") in the tree (i.e., random permutations always increased their distances). Nodes with low bootstrap support were collapsed prior to the analysis. A custom script for the computation of this index was written in R using the ape package for tree manipulations[70] and is available in Supplementary Note 1.

**Data availability.** The assemblies are available from EMBL or GenBank:
 PRJEB12080 GCA_900015485 *Microbotryum violaceum s. l.* from *Silene paradoxa* (1252) $a_2$
 PRJEB12080 GCA_900015495 *Microbotryum violaceum s. l.* from *Silene paradoxa* (1252) $a_1$
 PRJEB12080 GCA_900015415 *Microbotryum scabiosae* from *Knautia arvensis* (1118) $a_2$
 PRJEB12080 GCA_900008855 *Microbotryum scabiosae* from *Knautia arvensis* (1118) $a_1$
 PRJEB12080 GCA_900014955 *Microbotryum violaceum s. l.* from *Silene caroliniana* (1250) $a_2$
 PRJEB12080 GCA_900014965 *Microbotryum violaceum s. l.* from *Silene caroliniana* (1250) $a_1$
 PRJEB16741 ERZ348353 *Microbotryum silenes-acaulis* from *Silene acaulis* (1248) $a_1$
 PRJEB16741 ERZ348354 *Microbotryum silenes-acaulis* from *Silene acaulis* (1248) $a_2$
 PRJEB16741 ERP018599 Illumina genomes of *M. violaceum s. l.* from *S. caroliniana*, *S. virginica*, *S. paradoxa* and *M. Lagerheimii*

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

## Acknowledgements

This work was supported by the ERC starting grant GenomeFun 309403 to T.G., the Louis D. foundation award to T.G., the NSF DEB-1115765 and NIH R15GM119092 grants to M.E.H., the Marie Curie European grant 701646 to S.B., a postdoctoral fellowship (SFRH/BPD/79198/2011) from Fundação para a Ciência e a Tecnologia, Portugal to M.A.C, and DAAD, the Marie Curie European grant PRESTIGE-2016-4-0013 to F.H. and Campus France (PPP 57211753) to T.G. and D.B., and the Montana State University Agricultural Experiment Station. F.H. received the Young Biological Researcher Prize from the Fondation des Treilles, created by Anne Gruner Schlumberger, which supports research in Science and Art (http://www.les-treilles.com). We thank Cécile Fairhead for help with DNA extraction. PacBio sequencing was conducted at the IGM Genomics Center, University of California, San Diego, La Jolla, CA. We thank Jérôme Gouzy and Jean-Tristan Brandenburg for invaluable help with genomic analyses.

## Author contributions

T.G. and M.E.H. designed and supervised the study. T.G., M.E.H., D.B., and S.B. contributed to obtain funding. D.B. provided strains. H.B., M.E.H., S.L., A.S., and T.G. obtained the genomes. S.B., F.C., R.C.R.d.l.V., H.B., F.H., and M.A.C. performed the genomic analyses. D.M.d.V. performed analyses on gene genealogy allele clustering. T.G., S.B., and M.E.H. wrote the manuscript with contributions from all other authors.

## Additional information

**Competing interests:** The authors declare no competing interests.

