## [Peer Review File · Nature Communications]

Reviewers' comments:

Reviewer #1 (Remarks to the Author):

Branco et al. present an analysis of recombination on the mating-type chromosomes of anther-smut fungi. Mating-type chromosomes are analogous to sex chromosomes, except that the fungi lack the clear hallmarks of sexual dimorphism that typically are associated with sex chromosomes. The authors use gene order, evolutionary divergence and phylogenetic approaches to show that regions of recombination suppression have emerged around the mating type loci independently in multiple lineages. They go on to show that these regions bear the clear hallmarks of non-recombining regions (gene loss, transposon accumulation, etc) that have been shown on other nascent sex chromosome systems (including Ahmed et al. 2014 *Current Biology* 24: 1945-1957).

This manuscript appears to be largely an expanded version of recent work published by this group in PNAS (Branco et al. Evolutionary strata on young mating-type chromosomes despite the lack of sexual antagonism 114: 7067-7072). The case for complex inversions and strata of the mating-type chromosomes are displayed in Fig. 2 and Fig. 3 of the PNAS paper. I like the PNAS paper very much, and think that it presents a much-needed alternative to models of sex chromosome evolution. The work under consideration here includes additional species (there are five of the nine species in the paper under consideration here are also analysed in the PNAS paper), but uses the same methodological approaches and analyses, and comes to quite similar conclusions. I worry that the work under review here does not advance the story much beyond the PNAS paper, although this is ultimately up to the editor to decide.

I have no major quibbles with the paper beyond this concern about overlap with previous work.

Minor comments:

Line 82. Although I do agree with the authors that there have been few empirical tests, there is some emerging evidence for the theory of sexual conflict and recombination suppression, and even convergent evolution of sex chromosome strata across independent lineages (see Wright et al. Convergent recombination suppression suggests a role of sexual selection in guppy sex chromosome formation. *Nature Communications* 8: 14521).

Line 288. I am struggling to reconcile the authors' intended point. They argue that strata on the mating-type chromosome form in the absence of sexual conflict, and that therefore they are not sex-specific super-genes like Y or W chromosomes are thought to be. However, they also argue that the repeated evolution of these strata indicates strong selection for super-gene formation. What exactly might the function of these super-genes be? Also, there is an alternative, neutral model that there is simply a chromosomal breakpoint in the area. Breakpoints are known to be conserved across related species, leading to convergent inversions and fusions/fissions (e.g. Chutkar et al. *Genetics* 2008 179: 1657-1680). In the absence of clear phenotypes associated with the strata, it is difficult to make the case that they are indeed supergenes, and impossible to differentiate adaptive (supergene) from non-adaptive (conserved breakpoints) causes. I am not arguing that the data are inconsistent with the adaptive explanation offered, rather that the authors clarify that they cannot rule out the non-adaptive null hypothesis.

Reviewer #2 (Remarks to the Author):

Branco et al. have previously characterized the chromosome carrying the mating type loci of the anther-smut fungus *Microbotryum lychnidis-dioicae*, and found that it contains strata where recombination between the A1 and A2 alleles was suppressed at different time points. Here they extend this analysis to 4 new closely related fungi, and find that after independent chromosome rearrangements linked the PR and HR loci, a similar pattern can be found in two of them. The

recurrent linkage of HR and PR is very interesting. This is a great system in which to study the repression of recombination of mating types, and the detailed characterization provided here will be extremely useful.

*It is not entirely clear from parts of the text what is new (convergent linkage of PR and HR over a short period of time), confirmatory (appearance of strata after linkage of PR/HR), and previously reported (the possibility of strata in the absence of males and females).

For instance:

- in the abstract, the sentence "Anther-smut fungi lack male/female roles, showing that evolutionary strata can readily evolve without sexual antagonism, which stands in contrast with the current theory of sexual evolution." makes it sound like this is a novel finding of this paper, when it was reported by the same authors earlier this year.
- in the introduction, "we identified here five independent mating-type loci linkage events among eight *Microbotryum* species". I think that 4 new independent linkage events in each of the newly sequenced species is really what was identified.

I think this could be made more explicit.

*The analysis seems sound, but could have been better explained in the text:

- I don't think it is mentioned anywhere in the main text which of the genomes were obtained for this analysis.
- the "chromosome-length" assemblies turn out in some cases to yield 4 scaffolds per chromosome (I think the scaffolding based on centromeric repeats and ancestral gene order makes sense, but could be made clearer).
- in the supplementary methods they say that "In some cases, the center contigs could not be oriented as they did not include centromeric repeats", but don't give any more information about which species/contigs were affected.
- the date estimation was based on 17 genes that are in the light blue region in Figure 2. It would be useful to see those on Figure 3 as well.

In general I found it that not having a summary of the methods at all in the main text made it harder to follow the results.

*the PR and HD ancestral strata do not seem to consistently show very high D_s values: why is this?

*Several of the strata do not seem to significantly differ from the PAR in table S2. This should be mentioned (e.g. P10, first paragraph), as well as an explanation for how you determined the boundaries of this stratum.

*Results, P6: "alleles associated with the a1 mating type clustered together for both species": it would be useful to show gene trees for each of the 17 genes, as this would give a better sense of how consistent the patterns are.

*I think it is worth mentioning in P10 the evidence that the genes in the new non-recombining regions are "unrelated to mating-type".

Reviewer #3 (Remarks to the Author):

The manuscript by Branco, Giraud and co-workers addresses the evolution of mating type loci in smut fungi of the genus *Microbotryum*. The authors build their analyses on PacBio chromosome assemblies. Using genome comparisons, they reconstruct five independent rearrangements that all resulted in linkage of the mating type determining loci PR and HD. In this group of fungi, mating is predominantly selfing and linkage between the PR and HD loci is expected to be beneficial as it

increases mating compatibility between gametes. The authors thus conclude that these recent rearrangements in *Microbotryum* are the product of natural selection favoring linkage between the PR and HD loci. Based on gene genealogies, the authors infer the dates of rearrangements and the subsequent suppression of recombination.

The manuscript further explores the pattern of nucleotide divergence and reports the extension of recombination suppression from the mating type encoding genes into the surrounding chromosomal regions resulting in “evolutionary strata”, gene loss and TE accumulation.

Microbotryum serves as an excellent model to study the evolution of mating type loci in selfing species and the evolution of sex chromosomes. This study presents insight into the formation of non-recombining sex determining chromosome regions in multiple species of smut fungi. The identification of independent rearrangements in different smut species may provide an excellent example of convergent evolution. Some proof is still missing to make this final conclusion convincing.

A main concern with the present study is the absence of information about overall genome synteny between the five *Microbotryum* species. While, the study makes a strong case about evolution of the mating type loci by including multiple closely related species in their analyses, I miss to see a comparison of chromosome evolution at the mating type chromosome and the “autosomal” chromosomes. If structural rearrangements overall characterize genome evolution in *Microbotryum*, the linkage of the PR and HD loci may be the result of stochastic events that only subsequently have promoted selfing.

Likewise, it is not clear if the patterns of recombination suppression uniquely characterize the mating type locus or if similar patterns can be observed throughout the genome in other regions associated with rearrangements.

Trans-specific polymorphisms (and their absence) are used to make inference about the chromosomal fusion events. As mentioned above, the authors should provide information about the overall distribution of trans-specific polymorphisms to show that the pattern at the mating type locus differs from the remaining genome.

So essentially, the authors should integrate the analyses in a broader context of overall genome evolution for these species.

The same group of authors recently published a paper in PNAS (reference 5 in the submitted manuscript) about “Evolutionary strata on young mating-type chromosomes despite the lack of sexual antagonism”. This previous paper, also based on full-length chromosome assemblies, presents evidence of chromosomal fusion underlying the linkage of mating-type loci and it provides evidence, and dating, for evolutionary strata in the mating-type chromosome of *Microbotryum lychnidis-dioicae*.

In the current manuscript, the authors present evidence for additional independent events and the occurrence of similar evolutionary strata in other closely related smut species, and the manuscript confirms the same pattern observed in *M. lychnidis-dioicae*.

For the present manuscript, the authors have to strengthen the novelty of their findings. “More of the same” can be a strong argument to make conclusions about convergent evolution, but then it should be emphasized and additional analyses ideally included. This could be, as suggested above, a more detailed comparison of the mating type chromosomes and the “autosomal” chromosomes in the *Microbotryum* genus.

Additional comments:

L. 123: It is unclear how the “Genome comparison” is conducted, but if I understand correctly, it is based on blast analyses of the mating type loci? I recommend generating a genome alignment that will allow the authors to make a more detailed, and genome-wide, analyses of synteny and

gene order.

P. 5-9: More details about the number and predicted function (if available) of genes in the rearranged regions should be included in this characterization of the genomic regions involved in the re-arrangements.

L. 156: I am not sure how to interpret "chaotic rearrangements"

L. 203-205: This sentence is difficult to follow. Please re-phrase.

In the discussion I miss references to other fungal model systems, e.g. *Neurospora tetrasperma* in which similar patterns of rearrangements and recombination suppression are described.

L. 733-735: The analysis and dating of "recombination suppression" is not clearly explained. The authors use the divergence time of the alleles as the timing of the suppression of recombination. The authors should explain better the rationale behind this assumption.

Response to referee's comments:

Reviewer #1 (Remarks to the Author):

Branco et al. present an analysis of recombination on the mating-type chromosomes of anther-smut fungi. Mating-type chromosomes are analogous to sex chromosomes, except that the fungi lack the clear hallmarks of sexual dimorphism that typically are associated with sex chromosomes. The authors use gene order, evolutionary divergence and phylogenetic approaches to show that regions of recombination suppression have emerged around the mating type loci independently in multiple lineages. They go on to show that these regions bear the clear hallmarks of non-recombining regions (gene loss, transposon accumulation, etc) that have been shown on other nascent sex chromosome systems (including Ahmed et al. 2014 Current Biology 24: 1945-1957).

This manuscript appears to be largely an expanded version of recent work published by this group in PNAS (Branco et al. Evolutionary strata on young mating-type chromosomes despite the lack of sexual antagonism 114: 7067-7072). The case for complex inversions and strata of the mating-type chromosomes are displayed in Fig. 2 and Fig. 3 of the PNAS paper. I like the PNAS paper very much, and think that it presents a much-needed alternative to models of sex chromosome evolution. The work under consideration here includes additional species (there are five of the nine species in the paper under consideration here are also analysed in the PNAS paper), but uses the same methodological approaches and analyses, and comes to quite similar conclusions. I worry that the work under review here does not advance the story much beyond the PNAS paper, although this is ultimately up to the editor to decide.

>> We regret we were not clear enough in highlighting that it is the independent, repeated evolution of similar yet distinct chromosomal rearrangements for mating-type loci linkage that represents the important novelty here. Convergence is proposed as a fundamental pattern in evolutionary biology, showing that natural selection can independently produce similar phenotypes. This crucial question could not be addressed in the previous PNAS paper as we studied a single lineage. We understand the topic may have sounded similar to our previous paper, and we tried to better explain in the revised version that the novelty and importance resides in documenting independent and convergent evolution of similar beneficial supergenes. We have toned down the findings on evolutionary strata not involving mating-type genes and focused on the independent linkage of mating-type loci into beneficial supergenes, an aspect that was unfortunately unclear to Referee 1 and that we clarified.

I have no major quibbles with the paper beyond this concern about overlap with previous work.

>>We are glad to note that our conclusions and analyses are convincing.

Minor comments:

Line 82. Although I do agree with the authors that there have been few empirical tests, there is some emerging evidence for the theory of sexual conflict and recombination suppression, and even convergent evolution of sex chromosome strata across independent lineages (see Wright et al. Convergent recombination suppression suggests a role of sexual selection in guppy sex chromosome formation. Nature Communications 8: 14521).

>>We have toned down the evolutionary strata aspect to better highlight convergence of supergene evolution instead, and we no longer discuss the formation of evolutionary strata not involving mating-type genes, to focus on more novel aspects of the study.

Line 288. I am struggling to reconcile the authors' intended point. They argue that strata on the mating-type chromosome form in the absence of sexual conflict, and that therefore they are not sex-specific super-genes like Y or W chromosomes are thought to be. However, they also argue that the repeated evolution of these strata indicates strong selection for super-gene formation. What exactly might the function of these super-genes be? Also, there is an alternative, neutral model that there is simply a chromosomal breakpoint in the area. Breakpoints are known to be conserved across related species, leading to convergent inversions and fusions/fissions (e.g. Chutkar et al. Genetics 2008 179: 1657-1680). In the absence of clear phenotypes associated with the strata, it is difficult to make the case that they are indeed supergenes, and impossible to differentiate adaptive (supergene) from non-adaptive (conserved breakpoints) causes. I am not arguing that the data are inconsistent with the adaptive explanation offered, rather that the authors clarify that they cannot rule out the non-adaptive null hypothesis.

>>We are very sorry that our wording led Referee 1 to misunderstand this important point, and we tried to explain it better in the manuscript. There are several types of evolutionary strata in anther-smut fungi: some evolved for linking mating-type genes, representing adaptive supergenes, while others were subsequently generated without involving mating-type genes, therefore likely through neutral mechanisms. In our previous PNAS paper, we focused on the existence of multiple evolutionary strata not involving mating-type genes and we argued that this showed evolutionary strata can be formed without sexual antagonism (e.g. by chromosomal breakpoints as proposed by the Referee 1).

Here instead, we discovered the repeated formation of adaptive supergenes through chromosomal rearrangements linking the two fungal mating-type loci (HD and PR loci, controlling pre- and post-mating compatibility respectively). Such linkage of mating-type loci represents beneficial allelic combinations as it increases odds of gamete compatibility under selfing, as explained in figure S1. It is true and unfortunate that we put too much emphasis on the non-adaptive explanation for additional strata not involving mating-type genes in the previous version, something that was indeed already explained in our previous paper. In the revised manuscript, we have deleted most of such explanations and focus instead on the novel findings of convergence of adaptive supergenes, i.e., independent events of mating-type gene linkage. We also now show that autosomes remained highly collinear and without rearrangement within or across the species. The large-scale chromosomal rearrangements only involved PR and HD chromosomes and repeatedly led to regions of suppressed recombination with HD and PR loci as boundaries, indicating that the initial mating-type chromosome rearrangements were selected for linking HD and PR genes, forming adaptive supergenes.

Reviewer #2 (Remarks to the Author):

Branco et al. have previously characterized the chromosome carrying the mating type loci of the anther-smut fungus *Microbotryum lychnidis-dioicae*, and found that it contains strata where recombination between the A1 and A2 alleles was suppressed at different time points. Here they

extend this analysis to 4 new closely related fungi, and find that after independent chromosome rearrangements linked the PR and HR loci, a similar pattern can be found in two of them. The recurrent linkage of HR and PR is very interesting. This is a great system in which to study the repression of recombination of mating types, and the detailed characterization provided here will be extremely useful.

>>We thank Referee 2 for these positive comments, highlighting that the interest of this study was the independent linkage events of mating-type loci.

***It is not entirely clear from parts of the text what is new (convergent linkage of PR and HR over a short period of time), confirmatory (appearance of strata after linkage of PR/HR), and previously reported (the possibility of strata in the absence of males and females).**

>>We agree and we have revised the text to make it clear and highlight the important new findings.

For instance:

- in the abstract, the sentence "Anther-smut fungi lack male/female roles, showing that evolutionary strata can readily evolve without sexual antagonism, which stands in contrast with the current theory of sexual evolution." makes it sound like this is a novel finding of this paper, when it was reported by the same authors earlier this year.

- in the introduction, "we identified here five independent mating-type loci linkage events among eight Microbotryum species". I think that 4 new independent linkage events in each of the newly sequenced species is really what was identified.

I think this could be made more explicit.

>>We are glad the Referee 2 acknowledges the novelty of our findings (multiple convergent linkage events of PR and HD mating-type genes over a short period of time) and we have tried to better highlight the novel aspects and distinguish them from the more confirmatory findings (i.e. formation of subsequent evolutionary strata without sexual antagonism).

***The analysis seems sound, but could have been better explained in the text:**

- I don't think it is mentioned anywhere in the main text which of the genomes were obtained for this analysis.

>>We have clarified this point in the text and in Fig. 1 by asterisks.

- the "chromosome-length" assemblies turn out in some cases to yield 4 scaffolds per chromosome (I think the scaffolding based on centromeric repeats and ancestral gene order makes sense, but could be made clearer).

>>We have deleted the terms "chromosome-length".

- in the supplementary methods they say that "In some cases, the center contigs could not be oriented as they did not include centromeric repeats", but don't give any more information about

which species/contigs were affected.

>>We have clarified this point L515-516 (“The center contigs without centromeric repeats at any of their edges could not be oriented and were plotted in an arbitrary orientation”).

-the date estimation was based on 17 genes that are in the light blue region in Figure 2. It would be useful to see those on Figure 3 as well.

>>We have added the information on Figure 3. This made us realize the 17 genes were not all appropriate for inferring mating-type loci linkage in *M. silenes-acaulis* given its particular mating-type chromosome rearrangements. We therefore ran additional analyses on a restricted set of 13 genes and corrected the inferred mating-type loci linkage in *M. silenes-acaulis*. We thank the referee for this useful suggestion.

In general I found it that not having a summary of the methods at all in the main text made it harder to follow the results.

>>We have tried to better explain methods all along the result section.

***the PR and HD ancestral strata do not seem to consistently show very high Ds values: why is this?**

>>We have added the suggestion that it is likely due to occasional gene conversion events L294-295 (“Some of the genes in the ancient (blue and purple) strata had low dS levels in some species, probably due to occasional gene conversion events that reset the signal of divergence, as known to occur in fungal mating-type chromosomes”).

***Several of the strata do not seem to significantly differ from the PAR in table S2. This should be mentioned (e.g. P10, first paragraph), as well as an explanation for how you determined the boundaries of this stratum.**

>>We have clarified this point L298-299 (“The limit of the light-blue stratum was set at the most distal gene with a non-zero dS value, as all autosomes had zero dS values in the sequenced *M. v. caroliniana* individual (Supplementary Fig. 2a). [...] The mean dS value in this evolutionary stratum was not significantly different from that in the PARs (Supplementary Table 2), indicating that mating-type locus linkage was recent”). We have also added sequences and analyses on multiple genomes of *M. v. paradoxa* and *M. v. caroliniana* in order to further support our interpretation of recombination suppression in the young evolutionary strata.

***Results, P6: "alleles associated with the a1 mating type clustered together for both species": it would be useful to show gene trees for each of the 17 genes, as this would give a better sense of how consistent the patterns are.**

>>We have added a supplementary Figure showing the gene genealogies (Supplementary Fig. 3).

***I think it is worth mentioning in P10 the evidence that the genes in the new non-recombining**

regions are "unrelated to mating-type".

>>We have deleted this sentence as we focused less on this aspect in this manuscript.

Reviewer #3 (Remarks to the Author):

The manuscript by Branco, Giraud and co-workers addresses the evolution of mating type loci in smut fungi of the genus *Microbotryum*. The authors build their analyses on PacBio chromosome assemblies. Using genome comparisons, they reconstruct five independent rearrangements that all resulted in linkage of the mating type determining loci PR and HD. In this group of fungi, mating is predominantly selfing and linkage between the PR and HD loci is expected to be beneficial as it increases mating compatibility between gametes. The authors thus conclude that these recent rearrangements in *Microbotryum* are the product of natural selection favoring linkage between the PR and HD loci. Based on gene genealogies, the authors infer the dates of rearrangements and the subsequent suppression of recombination.

The manuscript further explores the pattern of nucleotide divergence and reports the extension of recombination suppression from the mating type encoding genes into the surrounding chromosomal regions resulting in "evolutionary strata", gene loss and TE accumulation.

Microbotryum serves as an excellent model to study the evolution of mating type loci in selfing species and the evolution of sex chromosomes. This study presents insight into the formation of non-recombining sex determining chromosome regions in multiple species of smut fungi. The identification of independent rearrangements in different smut species may provide an excellent example of convergent evolution.

>>We thank Referee 3 for these positive comments, also highlighting that the interest of this study was the independent linkage events of mating-type loci, revealing striking convergence.

Some proof is still missing to make this final conclusion convincing.

A main concern with the present study is the absence of information about overall genome synteny between the five *Microbotryum* species. While, the study makes a strong case about evolution of the mating type loci by including multiple closely related species in their analyses, I miss to see a comparison of chromosome evolution at the mating type chromosome and the "autosomal" chromosomes. If structural rearrangements overall characterize genome evolution in *Microbotryum*, the linkage of the PR and HD loci may be the result of stochastic events that only subsequently have promoted selfing. Likewise, it is not clear if the patterns of recombination suppression uniquely characterize the mating type locus or if similar patterns can be observed throughout the genome in other regions associated with rearrangements.

Trans-specific polymorphisms (and their absence) are used to make inference about the chromosomal fusion events. As mentioned above, the authors should provide information about the overall distribution of trans-specific polymorphisms to show that the pattern at the mating type locus differs from the remaining genome. So essentially, the authors should integrate the analyses in a broader context of overall genome evolution for these species.

>> We have added details and Figures (Supplementary Figs 2 and 4) illustrating genomic comparisons of autosomes, which show high collinearity within and between species, no or little differentiation

between mating types, and no trans-specific polymorphism. This indeed further shows that mating-type chromosomes have very particular evolution in terms of rearrangements and recombination suppression and increases the novelty compared to the previous paper; we thank Referee 3 for this interesting suggestion.

The same group of authors recently published a paper in PNAS (reference 5 in the submitted manuscript) about “Evolutionary strata on young mating-type chromosomes despite the lack of sexual antagonism”. This previous paper, also based on full-length chromosome assemblies, presents evidence of chromosomal fusion underlying the linkage of mating-type loci and it provides evidence, and dating, for evolutionary strata in the mating-type chromosome of *Microbotryum lychnidis-dioicae*.

In the current manuscript, the authors present evidence for additional independent events and the occurrence of similar evolutionary strata in other closely related smut species, and the manuscript confirms the same pattern observed in *M. lychnidis-dioicae*.

For the present manuscript, the authors have to strengthen the novelty of their findings. “More of the same” can be a strong argument to make conclusions about convergent evolution, but then it should be emphasized and additional analyses ideally included. This could be, as suggested above, a more detailed comparison of the mating type chromosomes and the “autosomal” chromosomes in the *Microbotryum* genus.

>>We have tried to better highlight the novelty of our work by reducing the text on evolutionary strata and focusing on convergence of mating-type loci linkage. We have also added autosomal comparisons (Supplementary Figs 2 and 4).

Additional comments:

L. 123: It is unclear how the “Genome comparison” is conducted, but if I understand correctly, it is based on blast analyses of the mating type loci? I recommend generating a genome alignment that will allow the authors to make a more detailed, and genome-wide, analyses of synteny and gene order.

>> We have clarified that we did perform genome-wide comparisons (L151-152: “Whole-genome BLAST comparisons revealed five different chromosomal rearrangements and fusions underlying the linkage between the HD and PR loci”), and we have added details and Figures (Supplementary Figs 2 and 4) illustrating genomic comparisons of autosomes, which show high collinearity within species and little rearrangements between species.

P. 5-9: More details about the number and predicted function (if available) of genes in the rearranged regions should be included in this characterization of the genomic regions involved in the re-arrangements.

>> We have reduced the part on evolutionary strata to focus on mating-type loci linkage and the list of functions in mating-type chromosomes has been published in the PNAS paper already.

L. 156: I am not sure how to interpret “chaotic rearrangements”

>> This term has been deleted.

L. 203-205: This sentence is difficult to follow. Please re-phrase.

>> We have clarified the sentence (L 280-281: "Such young evolutionary strata were defined as genomic regions with divergence between the alleles found in a1 genomes and those found in a2 genomes, but with lower levels of differentiation than for the genomic region ancestrally located between the PR and HD loci. We identified these young evolutionary strata by plotting dS levels between the alternate alleles along the inferred ancestral mating-type chromosome gene order").

In the discussion I miss references to other fungal model systems, e.g. *Neurospora tetrasperma* in which similar patterns of rearrangements and recombination suppression are described.

>> We have added references; note however that the situation in *N. tetrasperma* is different, as recombination suppression did not evolve for linking mating-type genes but instead the single mating-type locus to the centromere.

L. 733-735: The analysis and dating of "recombination suppression" is not clearly explained. The authors use the divergence time of the alleles as the timing of the suppression of recombination. The authors should explain better the rationale behind this assumption.

>> We have clarified the method and its rationale, both in the main text and in the material and methods.

References:

- Charlesworth, D. (2016). "The status of supergenes in the 21st century: recombination suppression in Batesian mimicry and sex chromosomes and other complex adaptations." *Evol. App.* 9(1): 74-90.
- Donoed et al., 2014 The coffee genome provides insight into the convergent evolution of caffeine biosynthesis. *Science* 345 : 1181-1184
- Duboue, E. R., et al. (2011). Evolutionary Convergence on Sleep Loss in Cavefish Populations. *Current Biology* 21(8): 671-676.
- Elmer, Kathryn R.; Fan, Shaohua; Kusche, Henrik; et al. 2014 Parallel evolution of Nicaraguan crater lake cichlid fishes via non-parallel routes *NATURE COMMUNICATIONS* 5: 5168
- Elmer, K. R. and A. Meyer (2011). "Adaptation in the age of ecological genomics: insights from parallelism and convergence." *Trends in Ecology & Evolution* 26(6): 298-306.
- Farhat MR, Shapiro BJ, Kieser KJ, et al. (2013) Genomic analysis identifies targets of convergent positive selection in drug-resistant *Mycobacterium tuberculosis*. *Nature Genetics* 45, 1183-U1320.
- Foote AD, Liu Y, Thomas GWC, et al. (2015) Convergent evolution of the genomes of marine mammals. *Nature Genetics* 47, 272-+.
- Gleiss, Adrian C.; Jorgensen, Salvador J.; Liebsch, Nikolai; et al. 2011 Convergent evolution in locomotory patterns of flying and swimming animals *NATURE COMMUNICATIONS* 2: 352
- Gould SJ 1989 *Wonderful Life – The Burgess Shale and the Nature of History*. Norton, New York.

- Haritos, Victoria S.; Horne, Irene; Damcevski, Katherine; et al. 2012 The convergent evolution of defensive polyacetylenic fatty acid biosynthesis genes in soldier beetles NATURE COMMUNICATIONS 3 : 1150
- Jensen, Niels Bjerg; Zagrobelny, Mika; Hjerno, Karin; et al. 2011 Convergent evolution in biosynthesis of cyanogenic defence compounds in plants and insects NATURE COMMUNICATIONS 2: 273
- Joron M, Frezal L, Jones RT, Chamberlain NL, Lee SF, Haag CR, Whibley A, Becuwe M, Baxter SW, Ferguson L, Wilkinson PA, Salazar C, Davidson C, Clark R, Quail MA, Beasley H, Glithero R, Lloyd C, Sims S, Jones MC, Rogers J, Jiggins CD, ffrench-Constant RH. (2011) Chromosomal rearrangements maintain a polymorphic supergene controlling butterfly mimicry. Nature 477(7363):203-6.
- Kapheim et al., 2015 Genomic signatures of evolutionary transitions from solitary to group living Science 348: 1139-1143
- Kowalko, J. E., et al. (2013). "Convergence in feeding posture occurs through different genetic loci in independently evolved cave populations of *Astyanax mexicanus*." Proceedings of the National Academy of Sciences of the United States of America **110**(42): 16933-16938.
- Kupper C, Stocks M, Risse JE, et al. (2016) A supergene determines highly divergent male reproductive morphs in the ruff. Nature Genetics 48, 79-+.
- Kwong, Waldan K.; Zheng, Hao; Moran, Nancy A. et al. 2017 Convergent evolution of a modified, acetate-driven TCA cycle in bacteria NATURE MICROBIOLOGY 2: 17067
- Lindgren, Johan; Sjovall, Peter; Carney, Ryan M.; et al. 2014 Skin pigmentation provides evidence of convergent melanism in extinct marine reptiles NATURE 506 : 484-+
- Mahajan S and Bachtrog D (2017) Convergent evolution of Y chromosome gene content in flies? NATURE COMMUNICATIONS 8: 785.
- Mahler DL, Ingram T, Revell LJ, Losos JB (2013) Exceptional Convergence on the Macroevolutionary Landscape in Island Lizard Radiations. Science 341, 292-295.
- Mani, Jan; Desy, Silvia; Niemann, Moritz; et al. 2015 Mitochondrial protein import receptors in Kinetoplastids reveal convergent evolution over large phylogenetic distances NATURE COMMUNICATIONS 6 : 6646
- Nagy, Laszlo G.; Ohm, Robin A.; Kovacs, Gabor M.; et al. 2014 Latent homology and convergent regulatory evolution underlies the repeated emergence of yeasts NATURE COMMUNICATIONS 5: 4471
- Parker, Joe; Tsagkogeorga, Georgia; Cotton, James A.; et al. 2013 Genome-wide signatures of convergent evolution in echolocating mammals NATURE 502 : 228-+
- Peichel CL (2017) Convergence and divergence in sex-chromosome evolution. Nature Genetics 49, 321-322.
- Pennisi E (2017) 'Supergenes' drive evolution. Science 357, 1083.
- Rubin, Benjamin E. R.; Moreau, Corrie S. 2016 Comparative genomics reveals convergent rates of evolution in ant-plant mutualisms NATURE COMMUNICATIONS 7 : 12679
- Purcell J, Brelsford A, Wurm Y, Perrin N, Chapuisat M (2014) Convergent genetic architecture underlies social organization in ants. Current Biology 24, 2728-2732.
- Seiffert, Erik R.; Perry, Jonathan M. G.; Simons, Elwyn L.; et al. 2009 Convergent evolution of anthropoid-like adaptations in Eocene adapiform primates NATURE 461 : 1118-U214
- Sherwood RK, Scaduto CM, Torres SE, Bennett RJ (2014) Convergent evolution of a fused sexual cycle promotes the haploid lifestyle. Nature 506, 387-+.

- Soria-Carrasco V, Gompert Z, Comeault AA, et al. (2014) Stick Insect Genomes Reveal Natural Selection's Role in Parallel Speciation. *Science* 344, 738-742.
- Stern DL (2013) The genetic causes of convergent evolution. *Nature Reviews Genetics* 14, 751-764.
- Steinmetz, Patrick R. H.; Kraus, Johanna E. M.; Larroux, Claire; et al. 2012 Independent evolution of striated muscles in cnidarians and bilaterians *NATURE* 487 : 231-U1508
- Storz JF (2016) Causes of molecular convergence and parallelism in protein evolution. *Nature Reviews Genetics* 17, 239-250.
- Thomas, Torsten; Moitinho-Silva, Lucas; Lurgi, Miguel; et al. 2016 Diversity, structure and convergent evolution of the global sponge microbiome *NATURE COMMUNICATIONS* 7 : 11870
- Wake DB, Wake MH, Specht CD (2011) Homoplasy: From Detecting Pattern to Determining Process and Mechanism of Evolution. *Science* 331, 1032-1035.
- Wang J, Wurm Y, Nipitwattanaphon M, et al. (2013) A Y-like social chromosome causes alternative colony organization in fire ants. *Nature* 493, 664-668.
- Wilson, Joseph S.; Williams, Kevin A.; Forister, Matthew L.; et al. 2012 Repeated evolution in overlapping mimicry rings among North American velvet ants *NATURE COMMUNICATIONS* 3: 1272
- Yeaman S et al. 2016 Convergent local adaptation to climate in distantly related conifers *Science* 353, 1431-1433
- Zhang et al., 2014 Comparative genomics reveals insights into avian genome evolution and adaptation *Science* 346: 1311-1320
- Zou, Zhengting; Zhang, Jianzhi 2016 Morphological and molecular convergences in mammalian phylogenetics. *NATURE COMMUNICATIONS* 7: 12758

Reviewers' comments:

Reviewer #2 (Remarks to the Author):

The authors have addressed my earlier concerns. The updated manuscript does a much better job at contrasting current and previous results.

Reviewer #3 (Remarks to the Author):

The authors show with an extensive genome dataset how independent rearrangements have caused the linkage of mating type loci across different smut fungi. My main concern with the manuscript still relates to the novelty of the study, in particular with respect to the PNAS paper Branco et al, 2017. The authors base their analyses on the same type of data including the distribution of trans-specific polymorphisms, synteny, dS and recombination suppression. The model presented in the current manuscript is likewise an extension of the mating-type chromosome evolution model shown in the PNAS paper. A main conclusion of the PNAS paper is the occurrence of multiple evolutionary strata, a finding that is also presented here although for mating type loci of different *Microbotryum* species.

The new data presented in the revised manuscript include further analyses of the autosomes to demonstrate the unique evolution of the mating type chromosomes.

I still find the summary of the whole genome analyses vague. For example, I could not find a proper summary of the whole genome assemblies showing the basic statistics. The data is only summarized in figure S2 and S4. In figure S2 we miss to see the actual contig length and gene number represented on the x-axis. In Figure S4 "well assembled contigs" were used to illustrate synteny between genomes. The less well- assembled contigs are likely those that bear the signatures of rearrangements and likewise will be enriched in TE. I would like to see a whole-genome-analyses/quantification of synteny. This could be represented by a mummer-plot or a circo plot giving an overview of the whole genome synteny conservation. We need to see the pattern at a genome scale and not only on the best-assembled contig. To fig S4 information about the numbers around the circo diagram are missing. I assume they indicate the length in Mb?

Centromeres are mentioned throughout the manuscript, and only at one place in the supplementary materials referred to as "putative centromeres". I could not find a reference to a study that has specifically identified and characterized the centromeres in *Microbotryum*. It is known that fungal centromeres can take very different form and range from point centromeres in yeast to long regional centromeres as found in *Neurospora*. If no other data available "putative centromeres" would be the right term to use throughout the text.

In the supplementary text I did not find much information about the clustering analyses used to generate the data presented in the boxplot in Fig S9. Please provide a more thorough explanation of the analysis.

Reviewers' comments:

Reviewer #2 (Remarks to the Author):

The authors have addressed my earlier concerns. The updated manuscript does a much better job at contrasting current and previous results.

>> We thank the referee for these positive comments and the help in improving the manuscript.

Reviewer #3 (Remarks to the Author):

The authors show with an extensive genome dataset how independent rearrangements have caused the linkage of mating type loci across different smut fungi. My main concern with the manuscript still relates to the novelty of the study, in particular with respect to the PNAS paper Branco et al, 2017. The authors base their analyses on the same type of data including the distribution of trans-specific polymorphisms, synteny, dS and recombination suppression. The model presented in the current manuscript is likewise an extension of the mating-type chromosome evolution model shown in the PNAS paper. A main conclusion of the PNAS paper is the occurrence of multiple evolutionary strata, a finding that is also presented here although for mating type loci of different *Microbotryum* species. The new data presented in the revised manuscript include further analyses of the autosomes to demonstrate the unique evolution of the mating type chromosomes.

>>As explained in our previous rebuttal letter and highlighted in the manuscript, the main result in this study is precisely the convergence of patterns, of both mating-type loci linkage and further evolutionary strata, that represents the novelty and striking pattern. The questions are very different from our previous paper, while methods are similar, but scientific questions matter for originality. Note that we also present novel results on degeneration levels and on autosomes comparisons, as acknowledged by the referee. We included some further data in the present revised version.

I still find the summary of the whole genome analyses vague. For example, I could not find a proper summary of the whole genome assemblies showing the basic statistics. The data is only summarized in figure S2 and S4.

>> We have added the information in the Supplementary Table 1.

In figure S2 we miss to see the actual contig length and gene number represented on the x-axis.

>> We have added the number of genes and the contig lengths in the legend.

In Figure S4 “well assembled contigs” were used to illustrate synteny between genomes. The less well- assembled contigs are likely those that bear the signatures of rearrangements and likewise will be enriched in TE. I would like to see a whole-genome-analyses/quantification of synteny. This could be represented by a mummer-plot or a

circo plot giving an overview of the whole genome synteny conservation. We need to see the pattern at a genome scale and not only on the best-assembled contig.

>> We have added a supplementary figure with whole genome circos comparisons (Supplementary Figure S4 i to n).

To fig S4 information about the numbers around the circo diagram are missing. I assume they indicate the length in Mb?

>> We have added this information in all figure legends (indeed length in megabases).

Centromeres are mentioned throughout the manuscript, and only at one place in the supplementary materials referred to as “putative centromeres”. I could not find a reference to a study that has specifically identified and characterized the centromeres in *Microbotryum*. It is know that fungal centromeres can take very different form and range from point centromeres in yeast to long regional centromeres as found in *Neurospora*. If no other data available “putative centromeres” would be the right term to use throughout the text.

>> We have added more details in the material and methods for explaining how we identified centromeres using a method specifically designed for this purpose and we have added information on the supplementary figure added (Supplementary Figure S4 i to n).

In the supplementary text I did not find much information about the clustering analyses used to generate the data presented in the boxplot in Fig S9. Please provide a more thorough explanation of the analysis.

>> The clustering index is described at the end of the material and method section, we have clarified this by adding a specific subheading and by referring to the material and method section in the legend of the supplementary figure 9. We have also added the script as a supplementary method.

REVIEWERS' COMMENTS:

Reviewer #3 (Remarks to the Author):

The authors have addressed the points raised in my previous review. Overall, I think they have improved the presentation of their results providing more convincingly the advance and novelty of their new findings.

I appreciate the additional analyses of synteny, and I only have one minor point: I am still not convinced about the definition of "centromeres" throughout the manuscript. As the specific repeats are not proven to be the centromeres, I still have a problem with the strong definition of centromeres used here. The correct term would be "putative centromeres". See Smith et al, 2012 Centromeres of filamentous fungi. Chromosom. Res. 20: 635–656.

We have made the last change asked by a referee, to replace “centromeres” by putative centromeres”.